# The No Free Lunch Theorem, Kolmogorov Complexity, and the Role of Inductive Biases in Machine Learning

## Abstract

No free lunch theorems for supervised learning state that no learner can solve all problems or that all learners achieve exactly the same accuracy on average over a uniform distribution on learning problems. Accordingly, these theorems are often referenced in support of the notion that individual problems require specially tailored inductive biases. While virtually all uniformly sampled datasets have high complexity, real-world problems disproportionately generate low-complexity data, and we argue that neural network models share this same preference, formalized using Kolmogorov complexity. Notably, we show that architectures designed for a particular domain, such as computer vision, can compress datasets on a variety of seemingly unrelated domains. Our experiments show that pre-trained and even randomly initialized language models prefer to generate low-complexity sequences. Whereas no free lunch theorems seemingly indicate that individual problems require specialized learners, we explain how tasks that often require human intervention such as picking an appropriately sized model when labeled data is scarce or plentiful can be automated into a single learning algorithm. These observations justify the trend in deep learning of unifying seemingly disparate problems with an increasingly small set of machine learning models.

## 1 Introduction

The problem of justifying inductive reasoning has challenged epistemologists since at least the 1700s (Hume, 1748). How can we justify our belief that patterns we observed previously are likely to continue into the future without appealing to this same inductive reasoning in a circular fashion? Nonetheless, we adopt inductive reasoning whenever we learn from past experience.

More recently, in the late 1990s, *no free lunch theorems* emerged from the computer science community as rigorous arguments for the impossibility of induction in contexts seemingly relevant to real machine learning problems (Wolpert, 1996; Wolpert & Macready, 1997). One such no free lunch theorem for supervised learning states that no single learner can achieve high accuracy on every problem (Shalev-Shwartz & Ben-David, 2014). Another states that every learner is equally good in expectation over a uniform distribution on learning problems (Wolpert, 1996). Such a world would be hostile to inductive reasoning. The assumption that labelings are drawn uniformly ensures that training data is uninformative about unseen samples.

In contrast to this dismal outlook on machine learning, naturally occurring data involve structure that could be shared even across seemingly disparate problems. If we can design learning algorithms with inductive biases that are aligned with this structure, then we may hope to perform inference on a wide range of problems. In this work, we explore the alignment between structure in real-world data and machine learning models through the lens of *Kolmogorov complexity*.

The Kolmogorov complexity of an output is defined as the length of the shortest program under a fixed language that produces it. In Section 3, we explain the connection between Kolmogorov complexity and compressibility. We note that virtually all data drawn from a uniform distribution as assumed by the no free lunch theorem of Wolpert (1996) cannot be significantly compressed, yet relevant real-world datasets are highly compressible. In particular, neural networks themselves can be used to create compressions of data labelings, upper bounding their Kolmogorov complexity.

We then demonstrate in Section 4 that modern neural networks also prefer low Kolmogorov complexity, complementing the low complexity of actual data. While models implemented on a computer cannot generate data with complexity exceeding the length of their associated program, we find they actually prefer data that is far simpler. We formulate simple languages for generating numerical sequences, under which we can directly measure the Kolmogorov complexity of a sequence. We use these languages to inspect the simplicity bias of both pre-trained and randomly initialized language models. GPT-3 (Brown et al., 2020) reliably favors less complex sequences, and bigger and better GPT-3 variants even more so. Notably, randomly initialized GPT models share this simplicity bias.

To further emphasize the universality of this simplicity bias, we reshape tabular data from diverse domains, including click prediction and airline delay prediction, into images and feed them through convolutional computer vision architectures, showing that these vision architectures prefer correct labelings to random ones, even on data which do not remotely resemble natural images and have no spatial structure. We then compute cross-domain generalization bounds via Kolmogorov complexity.

A common intuition associated with no free lunch theorems dictates that since a single learner cannot solve all problems, practitioners must inspect data and manually select an appropriate learner for the specific problem at hand. For example, a practitioner might select a more constrained model to avoid overfitting on small datasets, or convolutional architectures to accommodate natural image data. To the contrary, we show in Section 5 that the meta learner which selects the best learning algorithm from cross validation suffers little from overfitting even when the number of models investigated is large, and the cost of selection is quickly overcome by gains in validation accuracy.

Moreover, a single learner, which supports a variety of functions but prefers simple ones, can solve a wide range of problems. We show that flexible models accompanied by a penalty encouraging simple solutions can solve problems at a variety of sample sizes. In fact, the historic evolution of machine learning supports the ability of a single learner to perform diverse tasks (see Appendix A, Figure 4) as highly task-specific pre-neural algorithms, such as LDA (Blei et al., 2003) and HOG (Dalal & Triggs, 2005), were replaced by neural architectures such as convolutional or recurrent models, and transformers can now handily perform all tasks listed in Appendix A, Figure 4. We summarize our contributions as follows:

- We demonstrate the direct connection between compressibility and learnability that is implicit in no free lunch theorems by deriving a new no free lunch theorem using Kolmogorov complexity.
- We show that the low Kolmogorov complexity of real datasets can be directly derived from the machine learning models used to fit them.
- We compute the first cross-domain PAC-Bayes generalization bounds which show that neural networks such as convolutional architectures have low complexity biases that are relevant even on diverse tabular data far from what they were designed for.
- We demonstrate GPT-3's preference for sequences generated by short expression trees, and we find that even randomly initialized language models have a simplicity bias.

In short, while the no free lunch theorems are regularly used to justify the need for specially tailored inductive biases (Ho & Pepyne, 2002; Whitley & Watson, 2005; Ciuffo & Punzo, 2013; Watson et al., 1999), we show that real-world data are not only highly structured, but share their structure to a large extent. We further show how intervening to embrace a flexible hypothesis space together with a simplicity bias can lead to effective learners in small and large data regimes. Our findings explain recently observed phenomena, ranging from the generality of transformers, to the lack of overfitting on the test sets of popular benchmarks noted in Recht et al. (2019). We summarize several key takeaways throughout the paper in `blue`. In Appendix I, we provide an extended discussion.

## 2 BACKGROUND

We provide background on the no free lunch theorems, PAC-Bayes, and Kolmogorov complexity. We include an extended background discussion in Appendix B.

**No free lunch theorems.** No free lunch theorems (NFL) state that without making strong assumptions, a single algorithm cannot simultaneously solve all problems well. In supervised learning, the focus of this paper, Wolpert (1996) and Schaffer (1994) famously prove that every learner—a function that takes in labeled data and outputs a labeling function for the associated domain—achieves

the same average accuracy of $50\%$ on unseen examples over all binary classification problems. Shalev-Shwartz & Ben-David (2014) instead do not assume a particular distribution over learning problems and prove that for every learner, there exists a task on which the learner achieves poor accuracy with high probability over training splits, whereas another learner achieves perfect accuracy. Notably, the latter NFL computes accuracy over all data, not just "off-training" samples. The practical relevance of this theorem again hinges on the distribution over real-world learning problems and how well it aligns with the inductive bias of a learner. In this paper, we argue that the real-world learning problems we care about share a high degree of common structure, and the inductive biases of neural networks are well-aligned with such problems.

**Kolmogorov complexity and compression.** Kolmogorov complexity quantifies the structure in a bitstring, measuring the extent to which it can be compressed. For a fixed programming language $L$, the Kolmogorov complexity of data $x$, $K(x)$, is the length of the shortest program in that language that outputs $x$ (Kolmogorov, 1963). Analogous to conditional entropy, $K(y|x)$ is defined as the length of the shortest program which inputs $x$ and outputs $y$. Kolmogorov complexity provides a mathematical formalization of simplicity and Occam's razor, which encompasses many related concepts like Shannon information, compression, and minimum description length (MDL) (Li et al., 2008). While objects with large Kolmogorov complexity are impossible to verify (Chaitin, 1974), they are abundant over all possible bitstrings. All but exponentially few sequences of a given length have near maximal Kolmogorov complexity and are thus incompressible. Taken over the uniform distribution over bitstrings $x$, $P(K(x) \leq n - k) \leq 2^{1-k}$. However as we will discuss, these high complexity objects are extremely uncommon in practice.

**Universal induction.** Inspired by Kolmogorov complexity, a line of work introduces *universal induction* methods, which prefer low complexity answers (Solomonoff, 1964; Hutter, 2000; Lattimore & Hutter, 2013; Nakkiran, 2021). Notably, Solomonoff induction (Solomonoff, 1964; Rathmanner & Hutter, 2011) makes predictions by applying Bayes rule to the universal prior which favors low complexity, and provides learning guarantees. Rather than formalizing theoretical learners that rely on Kolmogorov complexity, which is in general uncomputable, Fernández-Delgado et al. (2014) and Gómez & Rojas (2016) test popular machine learning algorithms on a diverse array of datasets to see if any existing algorithms are plausibly universal. Another line of work shows that a single transformer model can perform well on many problems (Müller et al., 2022; Hollmann et al., 2022).

**PAC-Bayes generalization theory.** The PAC-Bayes framework is a convenient paradigm for proving generalization bounds on parametric models, while avoiding the pitfalls of uniform convergence. Rather than considering all elements of the hypothesis class on equal footing, we choose prior and posterior distributions over the parameters, and the generalization gap for elements of the posterior depends merely on the discrepancy between the two as measured by the KL divergence. This framework can explain many favorable properties of neural networks like flat minima (Hochreiter & Schmidhuber, 1997), noise resilience (Arora et al., 2018), and compressibility (Zhou et al., 2018). It can also provide nonvacuous generalization bounds, with recent bounds drawing directly from Kolmogorov complexity and the universal prior (Lotfi et al., 2022).

**On the relationship between our contributions and existing literature. (1)** In contrast to previous works which counter the no free lunch theorem by observing that a single model can achieve better-than-average empirical accuracy across diverse datasets (Fernández-Delgado et al., 2014; Gómez & Rojas, 2016), we explain and formalize the structures which are universal across such data distributions using Kolmogorov complexity. Relating this formalism to learning, we then show why low complexity is fundamental to such successes of machine learning models by proving a novel no free lunch theorem directly using Kolmogorov complexity. **(2)** The preference we demonstrate for low complexity emerges naturally in a variety of models, from transformer language models to convolutional neural networks, and requires no special interventions as proposed in Schmidhuber (1997) or Hinton & Van Camp (1993). **(3)** Existing generalization bound literature tunes priors on specific data distributions (Dziugaite & Roy, 2017; 2018; Pérez-Ortiz et al., 2021; Dziugaite et al., 2021) in line with the idea, often drawn from no free lunch theorems, that each domain requires a specially tailored model. In contrast, we demonstrate that neural networks can compress a wide range of datasets in domains they were not even designed for, and that this compressibility can explain generalization via PAC-Bayes generalization bounds. **(4)** Common wisdom dictates that neural network architectures must be carefully chosen for specific problems or sample sizes (Grinsztajn et al., 2022; Brigato & Iocchi, 2021; Lee et al., 2021), but we instead show through the formalism of complexity and experiments that specialized models can in principle be combined into a single learner which

can perform well on a wide variety of problems and sample sizes. Moreover, we show that the cost of model selection is minimal, explaining recently observed phenomena such as a lack of overfitting to the test sets of popular benchmarks (Recht et al., 2019).

## 3 UNPACKING THE NO FREE LUNCH THEOREM WITH KOLMOGOROV COMPLEXITY

The often cited no free lunch theorem of Wolpert (1996) states that all learners perform the same when averaged over a uniform distribution on all possible datasets. However, since most possible datasets are incompressible, the assumption of uniform samples subtly selects high complexity incompressible data, where learning is fundamentally impossible. We elucidate the centrality of complexity in NFL theorems by deriving a new NFL theorem which uses the incompressibility of random data to show why on this data learning is impossible. In Appendix C, we provide a brief introduction to bounding the Kolmogorov complexity of a dataset by compressing it and including the file sizes of both compressed file and decompression code. Through hypothesis testing, we rule out the possibility that real datasets are as high complexity as randomly drawn ones.

### 3.1 NNS AS COMPRESSORS OF THE LABELING FUNCTION

Relevant to supervised learning, we show that not only are unlabeled datasets compressible—the labeling functions are too. Further, we can demonstrate their compressibility concretely using trained models as compressors. Given a labeled dataset $\mathcal{D} = (X, Y) = \{(x_i, y_i)\}_{i=1}^n$, any likelihood model $p(y|x)$—regardless of whether the top predictions are correct—can be used to generate a lossless compression scheme to encode the dataset labels $Y$ given the inputs $X$. Using a stream code such as arithmetic coding (Witten et al., 1987), in combination with the probability model $p(y|x)$, the labels can be encoded in $K(Y|X, p) \leq -\sum_{i=1}^n \log_2 p(y_i|x_i) + 2$ bits (see e.g. MacKay (2003)). Models which maximize the log likelihood of the data also implicitly minimize the length of this encoding.

As we derive in Appendix D, $K(Y|X) \leq K(Y|X, p) + K(p) + 2\log_2 K(p) + c$, where $c$ is a small constant depending on the language. Writing the negative log likelihood in terms of the empirical cross entropy, combining our two inequalities, and dividing by the size of the dataset $n$ yields

$$\tfrac{1}{n}K(Y|X) \leq \frac{\text{CE}}{\ln 2} + n^{-1}(K(p) + 2\log_2 K(p) + c), \tag{1}$$

where CE is the cross entropy of the classifier $p$ averaged over dataset $\mathcal{D}$. This inequality implies that, regardless of how large the model is, it provides a non-trivial compression of the dataset as the size $n$ of the dataset grows sufficiently large, as long as CE is better than random guess. To demonstrate this fact, we employ the compression scheme from Lotfi et al. (2022) in order to find a compressed representation of MLPs on several class balanced tabular classification datasets (available at `openml.org`). As shown in Figure 1 (left), we are able to compress the labels on most of the datasets by well over the naive $n\log_2 C$ encoding length where $C$ is the number of classes. We also apply the method with convolutional architectures to compress labels on CIFAR-10 and CIFAR-100 in Figure 1 (middle), allowing us to reject the hypothesis that the labeling functions are drawn uniformly at random with extremely high confidence.

### 3.2 A KOLMOGOROV-STYLE NO FREE LUNCH THEOREM

A corollary of Equation 1 is that if the dataset is incompressible, then no model can do better than random chance in the large dataset limit, as we show in Theorem 1. Since compressible datasets in uniformly sampled data are exponentially unlikely, we can prove our own version of the no free lunch theoremWith very high probability, on any given uniformly sampled dataset, learning is impossible.

**Theorem 1.** *Let $(X, Y)$ be a labeled dataset with $n$ data points and uniformly sampled random labels from $C$ classes. Then, with probability at least $1 - \delta$, for every classifier $p(y|x)$,*

$$\text{CE}(p) \geq \ln C - \frac{\ln 2}{n}\left(K(p) + 2\log_2 K(p) + \log(1/\delta) + c\right), \tag{2}$$

*where CE(p) is the empirical cross entropy of the classifier $p(y|x)$ on the data. Thus for any model of bounded size, if the size of the dataset is large enough, the model cannot represent any classifier with cross entropy appreciably smaller than that attained from random guess. Proof found in Appendix D.*

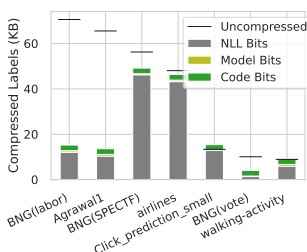 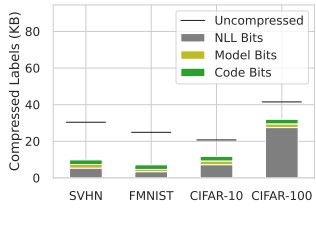 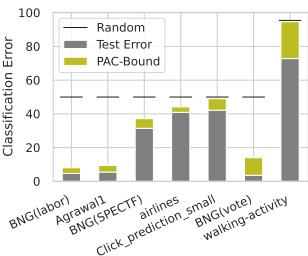

**Figure 1:** (Left): Compressed sizes of tabular labels where compression is performed via a trained MLP model (as in Section 3.1) vs. direct encoding of labels ($n \log_2 C$). (Middle): Compression of image classification datasets using CNNs. Note the breakdown of the total compressed size of the labels into model fit (NLL Bits), compressed parameters (Model Bits), and architecture and decompressor (Code Bits). In both cases, models can greatly compress a diverse suite of datasets, highlighting a common structure shared by models and real-world data. (Right): Compression based generalization bounds (Lotfi et al., 2022) for CNNs on tabular data, fed in with each pixel representing a tabular feature. The bounds are able to explain the majority of the model performance as shown by the test error, indicating that even CNNs designed for computer vision have a generic inductive bias appropriate for a wide range of datasets containing no spatial structure at all.

Like any of the no free lunch theorems, the necessary existence of unsolvable problems initially seems limiting. However, learning is in fact possible on compressible datasets (ones with less than maximal complexity).

> Real datasets are highly unlike the high complexity samples from the uniform distribution, associated with no free lunch theorems, where learning is impossible. The common structure shared by real datasets nullifies the limitations imposed by no free lunch theorems.

# 4 LOW-COMPLEXITY BIAS IN MACHINE LEARNING MODELS

Previously, we saw that real-world data distributions across domains share a common low Kolmogorov complexity bias. If we can construct models which prefer low-complexity data, we can hope to perform inference with a single model across many domains. While early machine learning systems incorporated highly domain-specific designs, such as handcrafted image features (Dalal & Triggs, 2005) or graphical models for language (Mnih & Hinton, 2007), modern neural network architectures across domains are converging on transformers (Vaswani et al., 2017; Dosovitskiy et al., 2020; Gulati et al., 2020; Somepalli et al., 2021), some of which can simultaneously achieve impressive performance on a variety of data types with a single architecture (Jaegle et al., 2021).

In this section, we argue that neural networks have a generic simplicity bias that extends beyond the datasets for which they are designed. To this end, we: (1) feed tabular datasets from diverse domains such as click prediction and airline delay prediction into convolutional networks designed specifically for computer vision and find that they provably generalize well due to their simplicity bias, (2) formulate a language with respect to which we can measure the Kolmogorov complexity of numerical sequences and observe that GPT-3 generates low-complexity sequences with exponentially higher probability, (3) predict the next term in a sequence with randomly initialized language models. Whereas the no free lunch theorem of Wolpert (1996) implies that such an inference procedure cannot outperform random guess on average, we find that randomly initialized neural networks prefer sequence completions which generate low-complexity completed sequences, demonstrating that they can make accurate guesses as long as the true sequence distribution also favors low complexity.

## 4.1 BOUNDING GENERALIZATION BY COMPLEXITY

Generalization bounds limit how the expected risk $R(h)$ for a model $h$ will differ from its train risk $\hat{R}(h)$. One simple such generalization bound is the finite hypothesis bound under a prior $P(h)$ (Langford & Seeger, 2001): with probability $1-\delta$: $R(h) \leq \hat{R}(h) + \sqrt{\frac{\log 1/P(h) + \log 1/\delta}{2n}}$. Relating to Occam's razor and Solomonoff induction, consider the universal prior that assigns higher likelihood to compressible hypotheses: $P(h) = 2^{-K_p(h)}/Z$ where $K_p(h) \leq K(h) + 2\log_2 K(h)$ is the *prefix*

Kolmogorov complexity and $Z \leq 1$. Combining the two, we have with probability $1 - \delta$,

$$R(h) \leq \hat{R}(h) + \sqrt{\frac{K_p(h) \log 2 + \log 1/\delta}{2n}}. \tag{3}$$

Despite the simplicity of the finite hypothesis bound, when combined with the universal prior, it provides nontrivial statements about generalization even for neural networks which have many more parameters than data points (Lotfi et al., 2022). Solutions found by many machine learning models on real datasets are highly compressible, and this reflects their bias for low Kolmogorov complexity functions. Even under an arbitrarily large or even infinite hypothesis space, generalization is possible if we assign prior mass disproportionately to the highly structured data that typically occurs.

### 4.2 NEURAL NETWORKS PREFER NATURALLY OCCURRING LABELINGS ACROSS DOMAINS

The inductive biases of even specialized architectures like convolutional networks facilitate broad learning abilities. We now illustrate how a preference for low complexity alone is sufficient for a high degree of generalization, provably, since real-world data labelings tend to have low complexity. To illustrate this fact, we take tabular classification datasets and encode the tabular features as an image by simply forming images where each pixel corresponds to a different feature, zero padding as necessary. We train a small convolutional network using this input data to predict the classification labels. Since the data has no local or translation equivariant structure, learning with the convolutional network requires overcoming its strong inductive bias which was hand tailored for settings with such structure. Even in spite of this extreme mismatch, the convolutional networks perform well. Using the compression and PAC-Bayes bound methodology from Lotfi et al. (2022) (see Equation 3), we show the generalization bounds on these models along with test error in Figure 1 (right). The strong generalization of convolutional networks on tabular datasets is almost entirely explainable through simplicity bias as shown by the fact that a finite hypothesis bound nearly matches the test error.

> Though CNNs were designed for vision, they generalize on unrelated tabular domains, a phenomenon almost entirely explained by their preference for low-complexity solutions.

### 4.3 GPT-3 ASSIGNS EXPONENTIALLY HIGHER PROBABILITY TO SIMPLER SEQUENCES

We now study the preference of GPT-3—a line of autoregressive LLMs—for simpler sequences. The ability of language models to solve reasoning problems has recently been studied by Zelikman et al. (2022), who develop a prompting framework, and d'Ascoli et al. (2022), who develop transformers for predicting symbolic expressions directly from the sequence. To perform our own study, we need a well-defined, computable notion of complexity. We thus define a simple, non-Turing-complete language and measure complexity with respect to this simple language. Namely, we generate integer sequences with binary expression trees. We then define the complexity of a sequence as the size of the smallest expression tree, measured by the number of internal nodes—or equivalently the number of operators in the expression represented by the tree—that generates that sequence. Note that while distinct from Kolmogorov complexity, the Kolmogorov complexity can be upper bounded by this complexity plus an added constant to encode the language. By using a small set of terms for the leaves and binary operators for the nodes, we can enumerate over all possible expression trees for small sizes at most $L$ and compute all sequences with complexities 0 through $L$.

In our experiments, we use operations $+$, $\times$, and $//$, where $//$ denotes integer division. For leaves, we use $2$ and $i$, where $i$ is the index within the sequence. For example, $(2 + i) \times i$ could be implemented with a tree of size 2 and would generate the sequence $a_i = 0, 3, 8, 15, ...$ Using this setup, we can generate sequences of varying complexity, according to a well-defined metric, and quantify the preference of GPT-3 models for simpler sequences over more complex ones. We provide details on how we tokenize sequences and extract their probabilities in Appendix F.

In Figure 2, we measure the average log-probability GPT-3 models assign to sequences of a given Kolmogorov complexity, where we fix the number of numerical tokens input into the model to be 30, and we observe that the probabilities assigned by these language models decrease exponentially with sequence complexity, similar to the Solomonoff prior discussed in Section 2. In contrast, a uniform prior would be described by a flat line. Since GPT-3 outputs per-token softmax probabilities conditional on all previous tokens within their context, we can compute the log-probability of

a sequence of tokens as $\log P(\text{Sequence}) = \sum_i \log P(\text{Token}_i | \{\text{Token}_{<i}\})$. Note we cannot easy measure the minimum description length of very complex sequences, so we limit our experiments to expression trees with at most 7 operators. In this low-complexity regime, we observe that big GPT-3 models which excel at language modeling, e.g. `Davinci` which contains 175 billion parameters, assign higher probability to these simple sequences than much smaller GPT-3 models such as `Ada`.

We can also examine the decay of such log-probabilities as we feed more tokens, corresponding to digits of sequence elements, into the model. As the sequences get longer, we see in Figure 2 that the probabilities assigned to sequences decay sub-exponentially, indicating that these models, especially bigger variants such as `Davinci`, become more and more confident about later sequence elements.

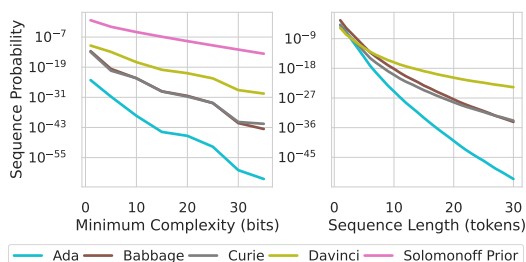

### 4.4 EVEN RANDOMLY INITIALIZED LANGUAGE MODELS PREFER LOW COMPLEXITY

The previous section examined pre-trained language models, but these models were trained on massive corpora of data. Do they prefer low complexity at initialization before they have even seen any data at all? While the initialization of neural network parameters is highly diffuse, these random parameters can induce a highly structured distribution over functions.

**Figure 2: GPT-3 prefers low-complexity sequences generated by expression trees.** **Left:** Average log-probability of sequences by complexity. **Right:** Average log-probability by sequence length, restricted to decimal digit tokens. GPT-3 variants ordered by increasing size. Observe that GPT-3 variants assign exponentially lower probabilities to higher complexity sequences (left), as in the Solomonoff prior, and bigger more powerful models especially exhibit this behavior. Moreover, the models become more confident as they see more tokens, and the more powerful GPT-3 variants such as `Davinci` learn faster (right).

Trained language models are known to repeat themselves (Holtzman et al., 2020; Fu et al., 2021). One might think that this behavior is learned from training data which contains repeated text, but we show that randomly initialized GPT models repeat themselves too. Interestingly, we can formalize the preference for repetition as a preference for low Kolmogorov complexity. In order to disentangle the impact of initialization from training, we adopt a simple language for generating binary sequences under which we can quickly measure Kolmogorov complexity. We consider a program to be a bitstring, and then the program upon execution simply repeats the bitstring until output reaches length 10. Under this language, the sequence $0, 0, 0, ...$ has Kolmogorov complexity 1, and $0, 1, 0, 1, ...$ has complexity 2, yet randomly generated sequences are exponentially more likely to have high complexity. We conduct our evaluations exhaustively on all such sequences of length 10.

We now generate sequences of length 10 with randomly initialized GPT-2 language models (Radford et al., 2019), using each initialization to generate one sequence, and we measure the frequency with which each sequence is generated. We estimate generation probabilities by Kolmogorov complexity in Appendix G where we see again that low-complexity sequences are assigned exponentially higher probabilities. Here, we compare (1) the uniform distribution over sequences, (2) randomly initialized GPT-2, as well as (3) pre-trained GPT-2 models. We see that randomly initialized parameters induce a structured distribution over sequences, and pre-trained checkpoints exhibit an even stronger preference for low complexity as they are trained on structured text. We can also use randomly initialized language models to perform next element prediction by estimating the probabilities they assign to the next element in a sequence given having correctly generated the previous terms. While Wolpert's no free lunch theorem (Wolpert, 1996) ensures that the average completion accuracy over all possible length 10 bitstrings is exactly 0.5, we verify in Appendix G that randomly initialized networks can be used for sequence completion when the sequence has low complexity.

We can further generate very long length-100 sequences with randomly initialized and pre-trained GPT-2 models and run a simple hypothesis test, demonstrating both randomly initialized and pre-trained models generate lower Kolmogorov complexity sequences on average than a uniform distribution. We generate 100,000 samples from each of these three generative distributions and perform a one-tailed t-test on the null hypothesis that $\mu(K(S_{\text{GPT}})) \geq \mu(K(S_{\mathcal{U}}))$, where $S_{\text{GPT}}$ and $S_{\mathcal{U}}$ respectively denote random sequences generated by the language model or a uniform distribution. Performing this hypothesis test, we reject this null hypothesis in both randomly initialized and pre-

trained models with an extremely low p-value, indicating that language models are indeed more likely to generate simple sequences. Details are found in Appendix G. We conclude that neural networks for language generation, both trained and randomly initialized, express a bias towards low Kolmogorov complexity which mirrors that of data as demonstrated in Section 3 and which was previously observed for classifiers in Valle-Perez et al. (2018).

> Language models, both pre-trained and randomly initialized, prefer to generate low-complexity sequences. As a result, we can use even such randomly initialized models to predict the next element in a sequence, as long as the sequence is low-complexity.

## 5    MODEL SELECTION WITH A SIMPLICITY BIAS

In typical industrial workflows, practitioners examine their data and select an appropriate learner. We can then consider the human model selector and the model they select as a single meta-learner. Whereas the no free lunch theorems seemingly preclude automated meta-learners which select performant models on any task, empirical works show that model selection can in fact be automated in practice (Vilalta & Drissi, 2002). Giraud-Carrier & Provost (2005) show that with minimal assumptions, the defeating conclusion of Wolpert's no free lunch theorem is escaped as long as datasets share structure so that the model selector generalizes to new datasets. In this section, we argue why in principle, model selection can be automated from the view of Kolmogorov complexity.

### 5.1    MODEL SELECTION AND GENERALIZATION BOUNDS

When developing a machine learning approach for an application, it is often helpful to leverage domain knowledge in constructing or choosing the right model for the task. One might start by choosing from families like MLPs, CNNs, GNNs, PointNets, or Transformers and then decide on the appropriate way of featurizing inputs, possibly incorporating knowledge of data symmetries via hard-coded equivariances or data augmentations. Even if we are extremely generous and suppose the practitioner is choosing from 100 million models, we can consider the impractical algorithm of selecting one via cross validation. While one might expect that such a procedure would overfit, even finite hypothesis bounds show that it does not. Using cross validation on a validation set of size 20000 for a classification problem, plugging in a uniform prior $P(h) = 1/|\mathcal{H}| = 10^{-8}$ to the finite hypothesis bound Equation 3, we get that the gap between validation and test error will be less than $3.4\%$ with probability greater than $99\%$. Ultimately, we avoid overfitting because we only need a number of data points proportional to the log of the size of the hypothesis space. This reasoning can also be applied to theoretically resolve the empirical observation in Recht et al. (2019) that we are not overfitting the test sets of popular benchmarks (more discussion in Appendix I).

For an even more general class of models, one may consider the number of bits needed to specify model architectures like MLPs, CNNs, or GNNs, as well as symmetries and any other required information. In each case, the architecture can be expressed in few bits. A near state-of-the-art computer vision model can be expressed in only 280 characters (Trockman & Kolter, 2022) in PyTorch. Similarly, important symmetries like translations, rotations, reflections, and other matrix groups can be expressed in few lines of code (Finzi et al., 2021b) and can be used to encode equivariances or for augmentation. Therefore, even in selecting from all possible models that can be expressed in that short amount of code, we can expect to generalize with only tens of thousands of data points.

> In principle, automating model selection directly via cross validation provably generalizes well across millions of models with only thousands of data points.

### 5.2    ONE MODEL FOR BIG AND SMALL TRAINING SETS

It is commonly believed that small training datasets demand compact architectures, whereas large training sets can accommodate flexible ones. Accordingly, practitioners hand select appropriate models for their datasets. We now show how we can intervene on the principle of combining flexibility with a simplicity bias, explored throughout the paper, to argue that a single learner can be

effective for all data sizes. Our prior should prefer simple functions we believe are more likely yet support a wide variety of functions. We begin with a simple illustration on polynomial regression.

**Polynomial regression.** Common intuition dictates that high degree polynomials overfit small training sets. In contrast, low degree polynomials cannot fit complicated functions so they should be avoided when training data is plentiful. However, we find that a single high degree polynomial can be effective across a wide variety of sample sizes as long as we encode a preference for low-complexity solutions, which rely on low degree coefficients. To this end, we adopt Tikhonov regularization with Tikhonov matrix $\mathrm{diag}(\{\alpha k^2\}_{k=0}^d)$; in particular, we impose an $\ell_2$ penalty that increases quadratically with the degree of the corresponding monomial. In Appendix H, we see that this model, which is flexible yet has a strong simplicity bias, performs at least on par with a low degree polynomial when training data is scarce, and with a high degree polynomial when training data is abundant.

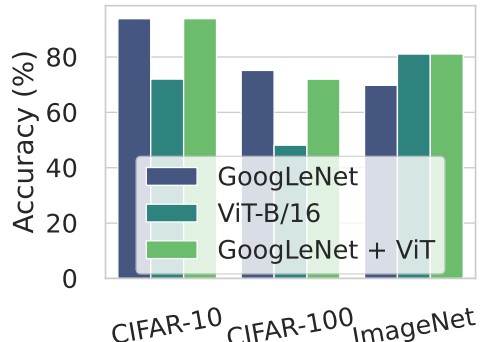

**Figure 3:** A single learner, which is more expressive than a ViT but also prefers simple solutions representable by a GoogLeNet, can simultaneously solve small and large scale problems.

**Neural networks.** We illustrate a similar concept with neural networks. We consider a small network, GoogLeNet (Szegedy et al., 2015), which performs well on small datasets such as CIFAR-10 and CIFAR-100 (Krizhevsky, 2009), but poorly on larger datasets like ImageNet (Deng et al., 2009). We also consider a large network, ViT-B/16 (Dosovitskiy et al., 2020), which performs significantly worse on CIFAR variants but better on ImageNet. As in the polynomial example, we can combine these two architectures, specifying our preference for GoogLeNet to the extent that it fits the training data. We train both models and then take a convex combination of their logits, $c *$ $\mathrm{logits}_{\mathrm{ViT}} + (1-c) * \mathrm{logits}_{\mathrm{G}}$, controlled by a parameter $c$ with $\ell_2$ regularization in favor of GoogLeNet (i.e., by adding $\lambda c^2$ to the loss function). In Figure 3, we observe that while GoogLeNet and ViT each have strengths and weaknesses, combining them with a preference for simplicity achieves the best of both worlds. While aggressively restricting our architectures can decrease computational cost, it is unnecessary for generalization. In other words, while GoogLeNet and ViT can be combined into a single learner with greater flexibility than GoogLeNet, and a stronger simplicity bias than ViT, so that manual selection between them is not required across data size.

In summary, flexible models with a low-complexity bias can be a one-stop-shop for machine learning since real-world data prefers low complexity. We do not need to compromise on flexibility in order to express a preference for low complexity solutions. Instead, follow Occam's Razor and choose the simplest explanation for the training set. We provide experimental details and additional experiments with Swin Transformer (Liu et al., 2021) in Appendix H.

> A single model can work well with both small and large training sets, so long as we embrace flexibility combined with a soft simplicity bias.

## 6 DISCUSSION

While large ML models are highly flexible, we saw that they reliably prefer low Kolmogorov complexity solutions—aligning well with relevant learning problems—despite not being designed with complexity in mind. This observation raises the question: why exactly do neural networks encode such a strong preference for low complexity and how can we tune this preference? Complementing the above observation, we also saw that a single expressive model which simultaneously supports a diversity of solutions but prefers simple ones can solve both simple and hard problems across sample sizes. Such learners present clear advantages over the current paradigm in deep learning in which we manually select small constrained architectures or large ones with mild inductive biases, depending on the problem. Keeping this possibility in mind, can we design expressive yet simplicity-biased models with affordable computational costs? We include an extended discussion of several fundamental themes that surface throughout the paper in Appendix I.

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

# A  INTRODUCTION FIGURE

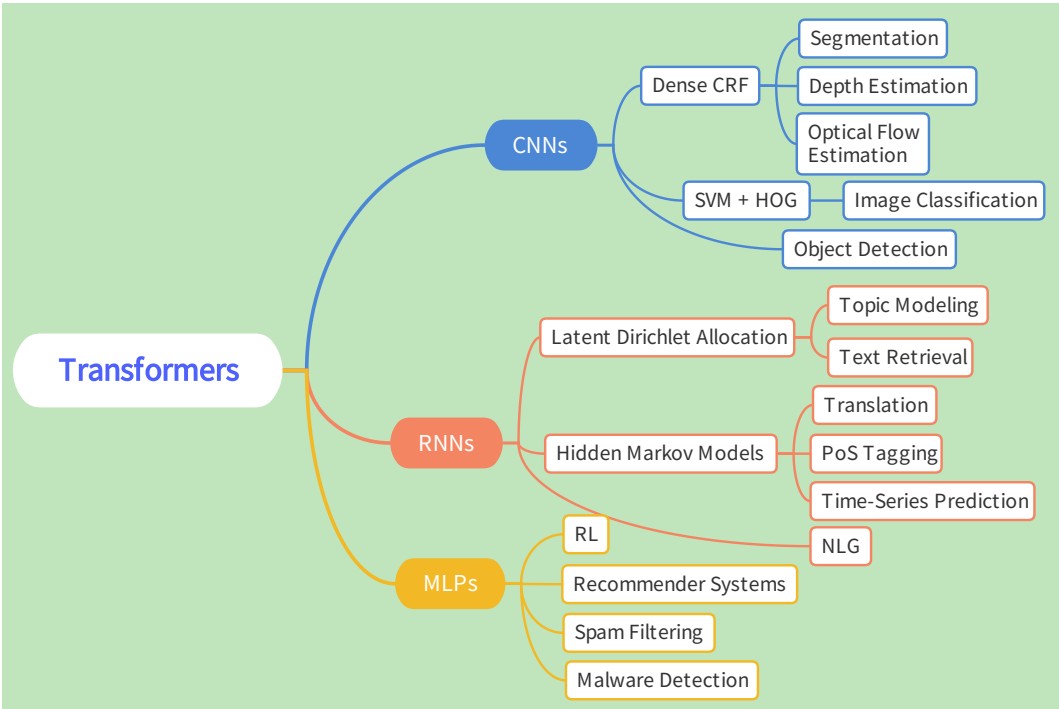

**Figure 4:** Over time, tasks that were performed by domain-specialized ML systems are increasingly performed by unified neural network architectures.

# B  EXTENDED BACKGROUND

This section provides an extended version of the background presented in Section 2.

**No free lunch theorems.** No free lunch theorems (NFL) state that without making strong assumptions, a single algorithm cannot simultaneously solve all problems well. No free lunch theorems for search and optimization indicate that all optimizers and search algorithms satisfying certain conditions perform exactly the same on average over all such search or optimization problems (Wolpert et al., 1995; Wolpert & Macready, 1997). In this work, we narrow our focus to NFL for supervised learning. Wolpert (1996), and similarly Schaffer (1994), proves an analogous theorem for supervised learning under which every learner—a function that takes in labeled data and outputs a labeling function for the associated domain—achieves exactly the same accuracy of $50\%$ on average over all binary classification problems where accuracy is only evaluated on unseen samples.

In order to prove no free lunch theorems, one needs to place very strong assumptions on the lack of structure in the labeling function, such as a uniform distribution, so that conditioning on the training labels does not modify the probability over labelings on unseen points (Rao et al., 1995). To illustrate the severity of this condition, imagine being presented a sequence of one million 1s and asked to predict whether the next element will be 1 or 0. If labelings of sequence elements were distributed uniformly, then we should assign equal probability to both options, even though intuition and Bayesian probabilistic models tell us overwhelmingly to favor 1.

Shalev-Shwartz & Ben-David (2014) instead do not assume a particular distribution over learning problems and prove that for every learner, there exists a task on which the learner achieves poor accuracy with high probability over training splits, whereas another learner achieves perfect accuracy. Notably, the latter NFL computes accuracy over all data, not just "off-training" samples. While this statement of NFL does not explicitly require uniformly distributed data, the existence of catastrophic failure modes for our learners would not matter if our learners never encountered them in

practice. After all, we do not care if our learners cannot solve problems we do not want to solve. Thus, the practical relevance of this theorem again hinges on the distribution over real-world learning problems and how well it aligns with the inductive bias of a learner. In this paper, we argue that the real-world learning problems we care about share a high degree of common structure, and the inductive biases of neural networks are well-aligned with such problems.

**Kolmogorov complexity and compression.** Kolmogorov complexity quantifies the structure in a bitstring, measuring the extent to which it can be compressed (an algorithmic definition of information content). For a fixed programming language $L$, the Kolmogorov complexity of data $x$, $K(x)$, is the length of the shortest program in that language that outputs $x$ (Kolmogorov, 1963). Analogous to conditional entropy, $K(y|x)$ is defined as the length of the shortest program which inputs $x$ and outputs $y$. Kolmogorov complexity provides a mathematical formalization of simplicity and Occam's razor, which encompasses many related concepts like Shannon information, compression, and minimum description length (MDL) (Li et al., 2008).

While objects with large Kolmogorov complexity are impossible to verify (Chaitin, 1974), they are abundant over all possible bitstrings. All but exponentially few sequences of a given length have near maximal Kolmogorov complexity and are thus incompressible. Taken over the uniform distribution over bitstrings $x$, $P(K(x) \leq n - k) \leq 2^{1-k}$, where $n$ denotes the string length. However as we will discuss, these high complexity objects are extremely uncommon in practice.

**Universal induction.** Inspired by Kolmogorov complexity, a line of work introduces *universal induction* methods, which prefer low complexity answers (Solomonoff, 1964; Hutter, 2000; Nakkiran, 2021). Notably, Solomonoff induction (Solomonoff, 1964; Rathmanner & Hutter, 2011) makes predictions by applying Bayes rule to the universal prior which favors low complexity, and provides learning guarantees. The existence of universal learners calls into question the broader message of no free lunch theorems, showing that Occam's razor or a preference for low-complexity data labelings is sufficient for learning on low-complexity data (Lattimore & Hutter, 2013).

Rather than formalizing theoretical learners that rely on Kolmogorov complexity, which is in general uncomputable, Fernández-Delgado et al. (2014) and Gómez & Rojas (2016) test popular machine learning algorithms on a diverse array of datasets to see if any existing algorithms are plausibly universal. These works indicate that Wolpert's no free lunch theorem (Wolpert, 1996) may not restrict machine learning in practice. Another line of work similarly shows that a single transformer model can perform well on a vast test bed of problems by mimicking Bayesian inference with in-context learning, and in fact achieves state-of-the-art on small tabular datasets (Müller et al., 2022; Hollmann et al., 2022).

**PAC-Bayes generalization theory.** The PAC-Bayes framework is a convenient paradigm for proving generalization bounds on parametric models, while avoiding the pitfalls of uniform convergence. Rather than considering all elements of the hypothesis class on equal footing, we choose prior and posterior distributions over the parameters, and the generalization gap for elements of the posterior depends merely on the discrepancy between the two as measured by the KL divergence. This framework can explain many favorable properties of neural networks like flat minima (Hochreiter & Schmidhuber, 1997), noise resilience (Arora et al., 2018), and compressibility (Zhou et al., 2018). It can also provide nonvacuous generalization bounds, with recent bounds drawing directly from Kolmogorov complexity and the universal prior (Lotfi et al., 2022).

Existing works in the generalization bound literature show that a model generalizes with high probability on a particular data distribution whenever it is compressible (i.e. has low complexity) with respect to the prior, but the prior is chosen specifically for the dataset at hand (e.g. CNNs for image classification) and furthermore the prior is often tuned directly on a fraction of the training set (Dziugaite & Roy, 2017; 2018; Pérez-Ortiz et al., 2021). In fact, there is a widely held belief in the generalization community that problem-specific priors, notably ones which are tuned on the training set, are necessary for strong generalization bounds (Dziugaite et al., 2021), and this belief manifests in data-dependent prior bounds across the literature.

In this paper, we argue against the necessity of problem-specific priors. Our generalization bounds and compression experiments show that a single low-complexity biased prior can suffice on a wide variety of problems due to the low Kolmogorov complexity of data. Whereas previous generalization theory literature is in line with the notion supported by no free lunch theorems that problems require specially tailored solutions, our work fights back against this widely held belief.

**Complexity in deep learning.** Several works have related Kolmogorov complexity to neural networks. One line of study proves that multi-layer perceptrons with Boolean input features are biased towards low-entropy functions, namely ones which classify disproportionately many or few points into the same class (Mingard et al., 2019) or are insensitive to flips in Boolean input features (De Palma et al., 2019). Pearlmutter & Rosenfeld (1990) argue that random initialization and noise in data increase the complexity of neural networks but that ensembling such models reduces complexity in expectation. Schmidhuber (1997) explicitly searches for simple neural networks with low Kolmogorov complexity and finds improvements in generalization on very small problems where such a search is computationally feasible.

The compressibility of neural networks has also been studied for purposes other than PAC-Bayes generalization bounds. For instance, Blier & Ollivier (2018) explore the compressibility of datasets using neural networks, and show that variational methods yield poor compression whereas prequential coding can be used to obtain shorter code lengths. Hinton & Van Camp (1993) find that adding noise to neural network weights during training reduces the information contained in the weights, getting bits back and making the parameter vector compressible. Our work shows that the compressibility of datasets using neural networks is universal and does not require domain-specific models.

**On the relationship between our contributions and existing literature.** We summarize the relationship between our contributions and existing literature as follows:

- In contrast to previous works which counter the no free lunch theorem by observing that a single model can achieve better-than-average empirical accuracy across diverse datasets (Fernández-Delgado et al., 2014; Gómez & Rojas, 2016), we explain and formalize the structures which are universal across such data distributions using Kolmogorov complexity. Relating this formalism to learning, we then show why low complexity is fundamental to such successes of machine learning models by proving a novel no free lunch theorem directly using Kolmogorov complexity.

- The preference we demonstrate for low complexity emerges naturally in a variety of models, from transformer language models to convolutional neural networks, and requires no special interventions as proposed in Schmidhuber (1997) or Hinton & Van Camp (1993).

- Existing generalization bound literature tunes priors on specific data distributions (Dziugaite & Roy, 2017; 2018; Pérez-Ortiz et al., 2021; Dziugaite et al., 2021) in line with the idea, often drawn from no free lunch theorems, that each domain requires a specially tailored model. In contrast, we demonstrate that neural networks can compress a wide range of datasets in domains they were not even designed for, and that this compressibility can explain generalization via PAC-Bayes generalization bounds.

- Common wisdom dictates that neural network architectures must be carefully chosen for specific problems or sample sizes (Grinsztajn et al., 2022; Brigato & Iocchi, 2021; Lee et al., 2021), but we instead show through the formalism of complexity and also through empirical experiments that specialized models can in principle be combined into a single learner which can perform well on a wide variety of problems and sample sizes. Moreover, we show that the cost of selection is minimal, explaining recently observed phenomena such as a lack of overfitting to the test sets of popular benchmarks (Recht et al., 2019).

## C  An Exercise in Bounding Dataset Complexity

We first consider the hypothesis that unlabeled machine learning datasets are drawn uniformly at random and use a bound on the Kolmogorov complexity as a test statistic. One can produce upper bounds on $K(x)$ by compressing $x$, but then it is necessary to include both the size of the compressed file and the size of the program required to decompress it. Alternatively, if one can construct a short program which directly outputs $x$, this program also forms a compression of $x$. Using `bzip2`, and including the size of the decompression program, we compress text dataset Amazon Review Full (McAuley & Leskovec, 2013) and audio dataset LibriSpeech (Panayotov et al., 2015) to 393.2 MB and 8.36 GB respectively, providing upper bounds on the Kolmogorov complexity with respect to the Python programming language. Computing the number of possible text and audio datasets of these sizes and supposing such datasets were in fact uniformly sampled at random, the probability of observing complexities this size or smaller is less than $10^{-1292913987}$ and $10^{-47632711550}$, astro-

nomically low p-values, conclusively ruling out the possibility that they were sampled uniformly in this way. If we randomly shuffle the datasets, we instead obtain bounds of only 836.7 MB and 9.69 GB, considerably larger, showing that the compressibility results not just from an inefficient encoding, but from structure in the dataset.

Other works have also examined Kolmogorov complexity in data, for example EEG patterns (Petrosian, 1995) or animal behavior (Zenil et al., 2015), and confirm that such data is simple. Our experiments above show that low Kolmogorov complexity is not specific to EEG patterns or animal behavior and is in fact a generic characteristic of common datasets we use in machine learning.

## D A KOLMOGOROV NO FREE LUNCH THEOREM PROOF

**Theorem 2.** *With probability at least $1 - \delta$ over datasets drawn from the uniform distribution, we have for every classifier $p$ over $C$ classes, the cross entropy is nearly as bad as random guess:*

$$\mathrm{CE}(p) \geq \ln C - \frac{\ln 2}{n} \left( K(p) + 2 \log_2 K(p) + \log_2 \delta + c \right)$$

.

*Proof.* Firstly, we relate the complexity of the classifier to the complexity of the labels $Y$ of the dataset:

$$K(Y|X) \leq K(Y, p|X) \tag{4}$$
$$K(Y|X) \leq K(Y|X, p) + K(p|X) + 2 \log_2 K(p|X) + c \tag{5}$$
$$K(Y|X) \leq K(Y|X, p) + K(p) + 2 \log_2 K(p) + c \tag{6}$$

For the second inequality, see e.g. (Fortnow, 2001).

We can bound $K(Y|X, p)$ by coding the labels using $p$

$$K(Y|X, p) \leq -\sum_{i=1}^{n} \log_2 p(y_i|x_i) + 2 \leq n\mathrm{CE}(p) + 2 \tag{7}$$

where $\mathrm{CE}(p)$ is the empirical cross entropy (see e.g. arithmetic coding in MacKay (2003)).

Rearranging, we have

$$\mathrm{CE}(p) \geq \frac{\ln 2}{n} \left( K(Y|X) - K(p) - 2 \log_2 K(p) - c \right).$$

Note that by simply counting all possible programs taking input $X$, there are less than $2^{k+1}$ labelings $Y$ with $K(Y|X) \leq k$. Note that there are $C^n$ distinct labelings, from which we are drawing uniformly. So that

$$\begin{aligned}
\mathbb{P}(K(Y|X) > n \log_2 C - m) &= 1 - \mathbb{P}(K(Y|X) \leq n \log_2 C - m) \\
&\geq 1 - \mathbb{P}(K(Y|X) \leq \lceil n \log_2 C \rceil - m) \\
&\geq 1 - \frac{2^{\lceil n \log_2 C \rceil - m + 1}}{C^n} \\
&\geq 1 - 2^{2-m}.
\end{aligned}$$

Alternatively, with probability at least $1 - \delta$,

$$K(Y|X) > n \log_2 C - \log_2 \frac{1}{\delta} - 3.$$

Therefore:

$$\mathrm{CE}(p) \geq \ln C - \frac{\ln 2}{n} \left( K(p) + 2 \log_2 K(p) + \log_2(1/\delta) + c \right)$$

$\square$

## E    PAC-Bayes Compression Experimental Details

For the OpenML tabular classification datasets, we preprocess them first by balancing the classes, subsampling all classes down to the number of examples of the least likely class. This way, when compressing the datasets, any result achieved is nontrivial in contrast with a very class imbalanced dataset. We heavily follow the compression method of (Lotfi et al., 2022), including the small 9 convolutional architecture which they use to generate their bounds. When cramming the tabular data into this convnet, we combine numerical features with one hot encoded categorical features and then pack these into the pixels of a 1 channel image, using however large an image as necessary to fit each of the different features inside.

With respect to the sizes of the random subspaces that we train the compressed models in, we consider 250,500,1000, and 2000. For tabular label compression, we employ a 2 hidden layer MLP with hidden dimension $k = 192$, and we consider the same 250,500,1000, and 2000 values for subspace dimension. We train for 80 epochs with 20 epochs of quantization at a batch size of 512 using Adam at lr= $3 \times 10^{-4}$. For image classification label compression, we use the 9-layer convnet with subspace dimensions 2000, 3000, 5000, and we train for 80 epochs using SGD at learning rate 0.1 and quantize for the remaining 20 epochs, at a batch size of 50. For calculating the length of the code for model architecture and decompressor, we need only the implementation of the model, the arithmetic decoder, and the loading of the quantized values. Removing wasted bits, we minified the python file, leading to a size of approximately 2.5KB.

## F    GPT-3 Experimental Details

To feed sequences into a model, we split up sequence elements into individual byte-pair encoding tokens corresponding to their decimal digits, and we place comma tokens between sequence elements as delimiters, also beginning every input with an `<|endoftext|>` token. We choose to use the byte-pair encoding of decimal digits with a space inserted before the digit, e.g. ' 0' as this is known to enhance the ability of language models to perform arithmetic (Zelikman et al., 2022). For example, the sequence $10, 11$ will be split up into ['`<|endoftext|>`', ' 1', ' 0', ',', ' 1', ' 1'], and each element of the list is tokenized individually. Then, the log-probability of a sequence is given by the sum of the log-probabilities corresponding to the correct decimal digits in their respective slots of the model's output. Note that various sequences will contain different numbers of decimal digits, and the sequence's log-probability will decrease with every token. Therefore, in order for fair comparison, we limit all sequences to 30 decimal digit tokens and truncate there.

## G    Sequence Generation and Completion with Randomly Initialized Language Models

For these experiments, we use Huggingface[1] GPT-2 architectures and pre-trained checkpoints. In order to estimate the probabilities assigned by randomly initialized language models to each bitstring, we generate one million random sequences, ensuring that many instances of each bitstring are generated as there are only $2^{10} = 1024$ bistrings of length 10.

We include here plots with the various sizes of GPT-2 architectures in Figure 5, Figure 6, and Figure 7.

We additionally include the hypothesis test referenced in Section 4.4. For this experiment, we generate 100,000 length-100 sequences from randomly intialized GPT-2 variants, pre-trained GPT-2 variants, and a uniform distribution. We then perform a one-sided t-test on the null hypothesis that $\mu(K(S_{\text{GPT}})) \geq \mu(K(S_{\mathcal{U}}))$, for both initialized and pre-trained models. Table 1 contains the resulting sample means, t-statistics and p-values. In all cases, we reject the null hypothesis with very low p-values, indicating that language models do prefer to generate low-complexity sequences. Notably, pre-trained language models exhibit an increased simplicity bias, and bigger and better language models even more so.

---

[1] https://huggingface.co/

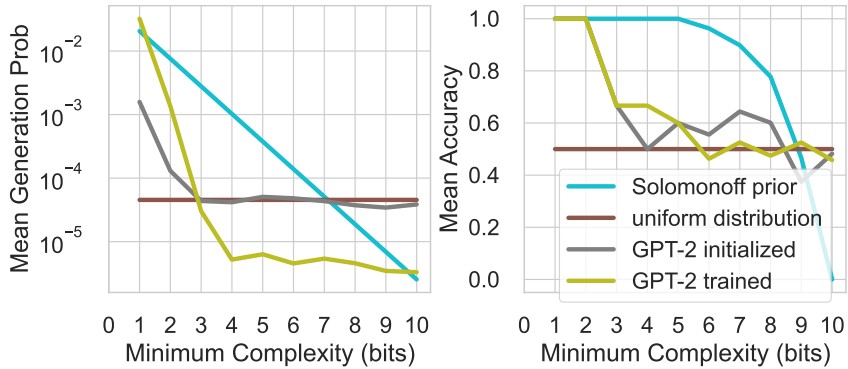

**Figure 5: Randomly initialized GPT-2 Base prefers low-complexity sequences generated by bitstring repetition. Left:** Average log-probability of sequences by complexity. **Right:** Average accuracy by complexity.

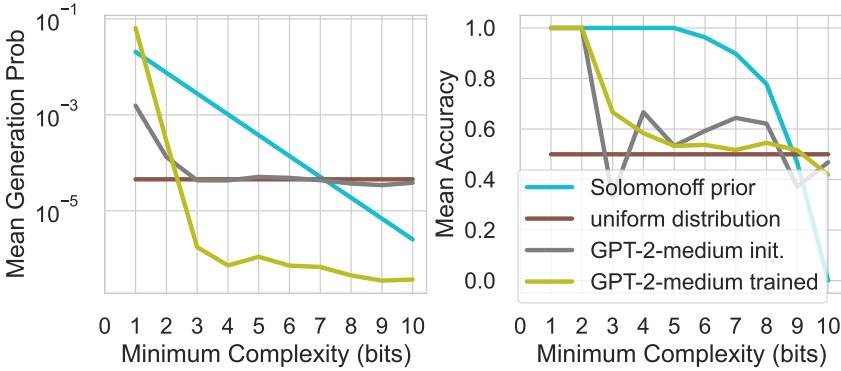

**Figure 6: Randomly initialized GPT-2 Medium prefers low-complexity sequences generated by bitstring repetition. Left:** Average log-probability of sequences by complexity. **Right:** Average accuracy.

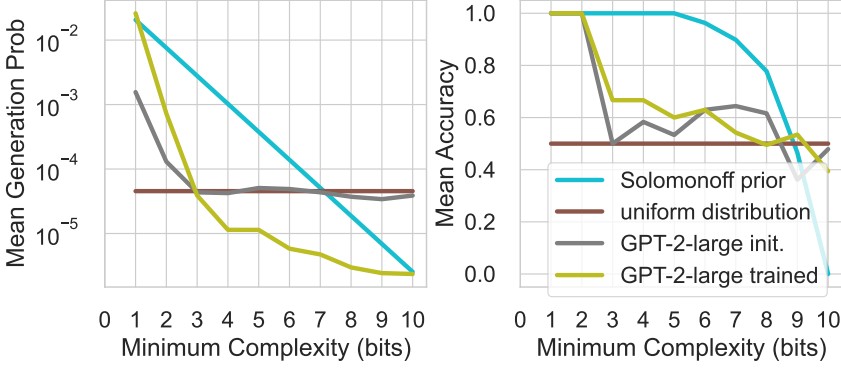

**Figure 7: Randomly initialized GPT-2 Large prefers low-complexity sequences generated by bitstring repetition. Left:** Average log-probability of sequences by complexity. **Right:** Average accuracy.

## H  BIG AND SMALL TRAINING SETS

**Polynomial regression.** We choose three example target functions on which to perform regression: $\cos(\frac{3\pi}{2}x)$, $x^2$, and $-36x + 49x^5 - 14x^7 + x^{10}$. Training data is randomly drawn from a uniform distribution over the unit interval, and we add noise to training labels from $\mathcal{N}(0, 0.1)$. In each case, for each dataset size, we average the mean squared error over 100 randomly sampled training sets. For Tikhonov regularized polynomial regression on the cosine and degree 2 polynomial target functions, we use $\alpha = 0.01$, and we use $\alpha = 0.001$ for regression on the degree 10 polynomial target function.

**Table 1: Hypothesis test for language model simplicity bias.** t-tests are one-sided, and p-values are rounded to 4 digits. We also report the mean Kolomogorov complexity of sequences generated by each language model and a uniform distribution.

| Model | $\overline{K(S_{\text{GPT}})}$ | t-statistic | p-value |
|---|---|---|---|
| Uniform Distribution | 98.36 | - | - |
| GPT-2 Base Initialized | 98.00 | -39.95 | 0.0000 |
| GPT-2 Medium Initialized | 97.99 | -40.91 | 0.0000 |
| GPT-2 Large Initialized | 98.00 | -40.11 | 0.0000 |
| GPT-2 Base Trained | 60.81 | -255.17 | 0.0000 |
| GPT-2 Medium Trained | 48.41 | -325.16 | 0.0000 |
| GPT-2 Large Trained | 46.34 | -342.80 | 0.0000 |

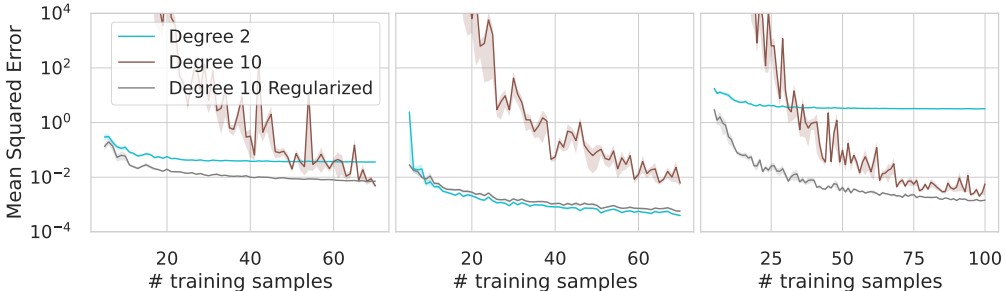

**Figure 8: High-order polynomials with a complexity penalty can solve problems at a variety of sample sizes.** **Left:** Cosine target function. **Middle:** Degree 2 polynomial target function. **Right:** Degree 10 polynomial target function.

**Image classification with neural networks.** For ImageNet trained models, we employ publicly available checkpoints from `torchvision`[2]. We train models on CIFAR-10 and CIFAR-100 for 200 epochs with initial learning rate 0.1 and cosine annealing along with horizontal flip and random crop augmentations. We use SGD with momentum 0.9 and batches of size 128. All CIFAR images are rescaled to $224 \times 224$ so that we can use an identical model for ImageNet and CIFAR data. In order to learn the parameter $c$ controlling the convex combination of models, we perform 10 epochs of training, where the models' parameters are frozen, and we apply weight decay with coefficient $10^{-5}$. We learn the parameter $c$ using SGD with momentum 0.9 and batch size 128, initial learning rate 0.1, and cosine annealing.

**Table 2:** Combinations of large and small architectures form single models that achieve high test accuracy on all dataset sizes. "GoogLeNet + ViT" denotes a model formed as a convex combination of the logits of the two constituent models with weight decay on the parameter $c$ controlling the convex combination which multiplies the logits of the larger model, ensuring that the small model is preferred as long as it fits the data.

| Model | CIFAR-10 | CIFAR-100 | ImageNet |
|---|---|---|---|
| GoogLeNet | 93.840 % | 75.160 % | 69.778 % |
| ViT-B/16 | 72.020 % | 48.140 % | 81.072 % |
| Swin-B | 74.710 % | 64.200 % | 83.582 % |
| GoogLeNet + ViT | 93.860 % | 71.990 % | 81.090 % |
| GoogLeNet + Swin | 93.760 % | 75.360 % | 83.150 % |

# I  EXTENDED DISCUSSION

In this section, we include an extended discussion of several fundamental themes that surface throughout the paper.

---

[2] https://pytorch.org/vision/stable/index.html

**Are the no free lunch theorems relevant to real-world model construction?** Not directly. They should not be invoked in discussions about the "need for strong inductive biases", or used as an argument that we cannot significantly automate machine learning or science, as they so often are. The assumptions of these theorems — such as datasets drawn from a uniform distribution over all datasets — are completely misaligned with the real world, where data are often highly structured, unlike uniformly sampled datasets (Section 3). There is a valid discussion to be had about the role of inductive biases in model construction, but the no free lunch theorems should play no part in this discussion. Our paper provides evidence, with PAC-Bayes bounds (Section 4.2), and generative likelihoods (Section 4.3 and Section 4.4), that the structure across many real-world datasets is shared to a surprising extent. These findings are aligned with the current movement towards similar transformer based architectures for many tasks, spanning vision, NLP, and tabular data, and away from more specialized models for each task (Figure 4).

**How do we build models that are broadly applicable?** Discovering the principles for building general purpose models is a fascinating research ambition. An emerging principle is that *we should embrace a flexible hypothesis space, while providing soft encouragement to learn salient structures in many real-world applications* (Wilson & Izmailov, 2020). In other words, we should leave behind restriction biases, often represented by hard architectural constraints, such as strict locality, parameter sharing, and translation equivariance, in favour of softer inductive biases. This principle is clearly exemplified by *residual pathway priors* (RPPs), which turn hard architectural constraints into soft inductive biases for equivariances (e.g., rotation or translation symmetries), efficiently guiding models towards structured solutions where that structure exists (Finzi et al., 2021a). These approaches have particularly strong performance where there are approximate equivariances, without suffering poor performance where there is not. Indeed, more generally, hard constraints are often unrealistic, and soft constraints in practice are sufficient for efficiently finding structure, while protecting against the misspecification and narrow applicability associated with the hard constraints.

We see this principle surfacing in many contexts. Transformers, for instance, lack hard constraints but have recently been found to discover more equivariant solutions even than models with hard constraints (Gruver et al., 2023). Outside deep learning, there is also work showing how soft inductive biases can be used to automate kernel selection on a variety of tasks which previously called for carefully hand-constructed kernels (Benton et al., 2019; Wilson & Adams, 2013; Lloyd et al., 2014). In this paper, we provide evidence that models can be made data efficient, while providing strong performance in larger data regimes, by embracing flexibility combined with soft inductive biases (Section 5).

**Should the model (i.e. learner) we use depend on how much training data are available? Should we use a different model on $n = 5$ points than $n = 10^6$ points?** The conventional wisdom is yes. We believe in principle, the answer is no. In practice, it depends.

- The *conventional wisdom* is that a model with a flexible hypothesis space cannot be well-determined from a small number of data points, and therefore will likely overfit and provide poor generalization. This conventional wisdom partly arises from early generalization theory, regarding Rademacher complexity (Mohri & Rostamizadeh, 2009), and VC dimension (Vapnik, 1998), as well as empirical overfitting of large models to small datasets. But there are now many empirical counterexamples to this thinking; indeed, we often use flexible models with many more parameters than datapoints, even without explicit regularization, and achieve good generalization (Zhang et al., 2016). It is not uncommon to train a neural network with millions of parameters on problems with tens of thousands of points, such as a ResNet-18 (He et al., 2016) on CIFAR-10 (Krizhevsky, 2009).

  But such counterexamples in fact long predate deep learning. For instance, Gaussian processes, which with standard kernels represent models with an infinite number of parameters, can perfectly fit most datasets. However, GPs rarely overfit and in fact generalize particularly well on small datasets (Rasmussen & Williams, 2006). Indeed, the examples of *benign overfitting* in *Understanding Deep Learning Requires Rethinking Generalization* (Zhang et al., 2016), where a convolutional net perfectly fits images with random labels, can be reproduced by several other model classes (Smith & Le, 2018; Wilson & Izmailov, 2020). Moreover, there are generalization theories, such as PAC-Bayes, which do not penalize large hypothesis spaces, but instead focus on which solutions are a priori *likely* under the model, rather than which solutions are expressible (McAllester, 1998; Alquier, 2021).

- *In principle*, our beliefs about the generative process for data would not typically depend on how many points we happen to observe (Neal, 1996). Moreover, honestly representing our beliefs should not lead to poor generalization in an ideal modeling paradigm. Typically we would believe that there are many solutions that are a priori *possible*, even if most of them are not a priori *likely*. We should therefore embrace a large hypothesis space, regardless of how much data we have access to (Wilson & Izmailov, 2020). While a small model may be quickly constrained by the data, it will be erroneously constrained, which can be particularly problematic in representing epistemic uncertainty (which solutions are possible given limited information) (MacKay, 1995). We have shown several examples of how we can construct models that are competitive in both small and large data regimes, by embracing the principle of a flexible hypothesis space combined with soft inductive biases towards simplicity.

- *In practice*, it can sometimes be computationally cumbersome to work with large models (or learners incorporating multiple architectures as in Section 5.1) on small data, such that it is not worth the computational trade-offs. It can also be more convenient to take a "brute force" approach, manually scaling models with more data points. But if one is willing to put the necessary care into representing our honest assumptions, then a single model approach is likely to provide better performance across different data regimes. We have seen here how this single model approach can be possible, by combining flexibility with soft simplicity biases (Section 5.2).

**How crucial are the implicit biases of the optimization procedure in finding simple generalizable solutions?** The ability for deep models to generalize is often attributed to the implicit biases of stochastic optimizers like SGD or Adam (Amir et al., 2021; Wu et al., 2020). However, good generalization can be achieved with full-batch training procedures, even in the absence of explicit regularization (Geiping et al., 2021; Izmailov et al., 2021) or without gradient-based optimization at all (Chiang et al., 2023). Indeed, while stochastic optimizers can sometimes confer a small gain in performance, it is largely the design of the architecture that makes generalizable solutions more easily accessible (occupy a greater volume in the loss landscape), rather than the optimizer selecting for particularly generalizable low-loss solutions. For example, as we increase the dimensionality of the parameter space, flat solutions occupy an exponentially increasing volume, and these solutions often provide more compressed representations of data, typically leading to better generalization (Hochreiter & Schmidhuber, 1997; Kawaguchi et al., 2017; Huang et al., 2020; Maddox et al., 2020; Foret et al., 2020). In Bayesian terms, the posterior volume of low-loss solutions that provide good generalization is much larger than the volume of low-loss solutions that provide poor generalization, as a consequence of the architectural prior (Wilson & Izmailov, 2020).

**Should we be worried that we are inadvertently overfitting to popular benchmarks by comparing so many models on these benchmarks?** Model development has been motivated to a large extent by improving test-set performance on popular benchmarks, in a sense "training on test", leading to a concern that we may be overfitting to particular test sets. Indeed, this concern is the catalyst of the study *"Do ImageNet Classifiers Generalize to ImageNet?"* (Recht et al., 2019), where new test sets for CIFAR and ImageNet are created because "the research community could easily be designing models that only work well on the specific test set but actually fail to generalize to new data" (Recht et al., 2018).

This study finds that the rankings of the best performing models is essentially preserved, concluding that drops in performance are likely due to minor distribution shifts in the new test sets, rather than overfitting.

While this finding is often characterized as surprising, our calculation in Section 5.1 provides a theoretical explanation for these results. Even comparing one hundred million models on a test set smaller than CIFAR or ImageNet, we would not expect a drop in test error by more than a few percent.

In short, the danger of overfitting to standard benchmarks by checking the test accuracy of many different models is in fact provably very small.

**Are recent demonstrations of GPT and large language models surprising?** Large language models have recently shown versatility, capable of problem solving in a wide variety of contexts, ranging from passing the bar exam (Katz et al., 2023), university-level physics or chemistry prob-

lems (Lewkowycz et al., 2022), or even general-purpose reasoning in science and math (Taylor et al., 2022). Part of the reason these demonstrations are perceived as surprising is due to the influence of the no free lunch theorems, which imply one model can't effectively solve many tasks. We have shown how this intuition is misguided. Real-world modeling problems are often highly structured, and that structure is shared across problems to a large extent. We have also demonstrated that GPT-3 and GPT-2 models, even randomly initialized, have a simplicity bias. This simplicity bias, in combination with their flexibility, makes large language models good candidates as general-purpose problem solvers, though of course they will not be a magic bullet for every problem, or without limitations.

## J    LIMITATIONS

In multiple experiments in this paper, we bound Kolmogorov complexity by compressing datasets or models. These upper bounds on Kolmogorov complexity are likely very loose since our compressions probably are far from optimal. We expect that high-performance models and the distribution over real-world datasets possess a much more severe simplicity bias than we can prove. Moreover, a number of the experiments in this paper should be viewed as proof of concept. For example, the compressibility of various datasets using neural networks provides evidence that real-world datasets are highly non-uniform and share a generic structure in common with neural networks. However, we can't be sure that these observations will hold for all datasets and models as facts about the distribution of real-world datasets cannot be proven mathematically.

## K    COMPUTATIONAL RESOURCES

In order to perform all experiments in this paper, we used a total of approximately 200 GPU hours on NVIDIA RTX A4000 and NVIDIA Titan RTX cards.

