# OpenReview forum: "The No Free Lunch Theorem, Kolmogorov Complexity, and the Role of Inductive Biases in Machine Learning"
_ICLR.cc/2024/Conference — Submitted to ICLR 2024_

### Official Review · Reviewer_hV8P · 2023-10-24

**Soundness:** 4 excellent
**Presentation:** 4 excellent
**Contribution:** 3 good
**Rating:** 6
**Confidence:** 5

**Summary:**

This research delves into the importance of inductive biases within the field of machine learning, specifically by examining the concept of Kolmogorov complexity. The study conducts experiments and uncovers several noteworthy findings: 1) real-world datasets tend to have low complexity, 2) convolutional neural networks can efficiently compress tabular data, 3) language models at initialization tend to favor sequences with low complexity, and 4) model selection can be automated via cross-validation and provably generalizes. These findings support the idea that using large and adaptable models can yield benefits in a variety of tasks without the need for additional inductive biases provided by humans.

**Strengths:**

**Originality**

 This paper stands out for its unique approach, synthesizing various observations to construct a compelling argument that challenges the necessity of specialized inductive biases in solving practical machine-learning tasks. Although some of the experimental results might not be groundbreaking, the amalgamation of these findings into a coherent argument is a novel contribution.

**Quality**

The paper impressively showcases well-motivated analyses and experiments, exhibiting a high level of technical rigor. Moreover, the analyses and experiments are discussed and interpreted extensively, which is critical for a paper of this kind.

**Clarity**

Notably well-written, the paper benefits from clear references to previous work, enhancing its readability, as well as a strong overall structure. The inclusion of takeaway boxes and an extensive discussion in the Appendix further adds to its clarity and comprehensibility.

**Significance**

The paper addresses a substantially significant question in the field of machine learning, namely the necessity of specialized inductive biases. This question, often overlooked, receives a strong affirmative response in this paper, backed by ample evidence. Therefore, the paper has the potential to make a notable impact on the research community. This is particularly true given the prevalence of the alternative viewpoint: that specific inductive biases are necessary to achieve efficient generalization on most practical problems.

**Weaknesses:**

The primary weakness I perceive in the paper is that its technical findings, although robust, may not be particularly surprising. Although the specific analyses in the paper are new, the paper reiterates established facts, such as the structured nature of real-world datasets, neural networks' preferences for simpler solutions, and the automation of model selection. While the paper's innovation lies in its synthesis of these facts, the lack of groundbreaking technical discoveries is nevertheless a weakness.

**Questions:**

The paper is quite well written and complete; as a result, I have no technical questions. I hope the authors are able to emphasize the significance of the contributions in this work as mentioned in the weaknesses section.

---

> ### Author Response · Authors · 2023-11-19
> **Author Response to Reviewer hV8P 1/2**
>
> Thank you for your supportive feedback and for recognizing the significance, quality, and originality of our work.  As you point out, the compressibility of real-world data has long been studied by information theorists, and the consequent benefits of simplicity bias have historically been argued by theoretical computer scientists, especially in the wake of early no free lunch theorems.  In our paper (Appendix B), we also reference literature on various notions of simplicity bias applied to neural networks.  In this context, our work is valuable in several ways:
>
> - The story we synthesize in our work and support with empirical evidence, namely that the low-complexity bias of neural networks enables general-purpose learners due to the shared structure underlying real world data, is actually not widely accepted across the deep learning community.  We often see papers or prominent members of the ML community arguing for the fundamental necessity of problem-specific learners, sometimes explicitly referencing no free lunch theorems.  For example, [1] say “the only way one strategy can outperform another is if it is specialized to the specific problem under consideration”, [2] reference the NFL in stating that “the business of developing search algorithms is one of building special-purpose methods to solve application-specific problems”, [3] say “the performance of the different algorithms over the 42 experiments considerably differ. This proves the validity of NFL theorems in our field”, and [4] reference NFL to explain why success on synthetic problems may not transfer to real ones.  See [5,6] for examples of prominent community members leveraging the NFL to argue for specialized learners.  Our work argues against these views intuitively, theoretically, and empirically.
> - We present a number of novel experiments which we think will be interesting to the broad machine learning community, for example understanding the preferences of LLMs or repurposing tools from PAC-Bayes generalization theory to understand structure in data and to show that seemingly specialized models like CNNs actually possess a generic simplicity bias that allows them to compress and provably generalize on tabular datasets for which they were not designed.
> - We phrase the no free lunch problem for machine learning as one of compression, which allows us to directly delineate between the no free lunch regime where data is incompressible, making learning impossible, and the regime which we actually observe in real world datasets, namely where data is highly structured and in fact shares structure in common with machine learning models, for example as measured by compressing labeling functions using neural networks which allows for highly nonvacuous PAC-Bayes generalization bounds.

---

> > ### Author Response · Authors · 2023-11-19
> > **Author Response to Reviewer hV8P 2/2**
> >
> > On top of the above points, we want to touch on a crucial distinction between our work and existing generalization bound literature.  Existing works, many of which we cite in our submission, show that a model generalizes with high probability on a particular data distribution whenever it is compressible (has low complexity) with respect to the prior, but the prior is chosen specifically for the dataset at hand (e.g. CNNs for image classification) and furthermore the prior is often tuned directly on the training set [7-9].  In fact, there exists a widely held belief in the generalization community that problem-specific priors, notably ones which are tuned on the training set, are necessary for strong generalization bounds [10], and this belief manifests in data-dependent prior bounds across the literature.
> >
> > In stark contrast, the necessity of problem-specific priors is exactly what we argue against in our paper.  Our generalization bounds and compression experiments show that a single low-complexity biased prior can suffice on a wide variety of problems due to the low Kolmogorov complexity of data.  Whereas previous generalization theory literature is in line with the notion supported by no free lunch theorems that problems require specially tailored solutions, our work fights back against this widely held belief.
> >
> > Thank you again for your thoughtful review. We made a significant effort to address your feedback and would appreciate it if you would consider raising your score in light of our response.  Please let us know if you have additional questions we can address.
> >
> > **References:**
> > [1] Ho, Yu-Chi, and David L. Pepyne. “Simple explanation of the no-free-lunch theorem and its implications.” *Journal of optimization theory and applications* (2002).
> > [2] Whitley D, Watson JP. “Complexity theory and the no free lunch theorem.” *Search methodologies* (2005).
> > [3] Ciuffo, Biagio and Vincenzo Punzo. "“No free lunch” theorems applied to the calibration of traffic simulation models." *IEEE Transactions on Intelligent Transportation Systems* (2013).
> > [4] Watson J-P, Barbulescu L, Howe AE, Whitley LD. Algorithm performance and problem structure for flow-shop scheduling. *AAAI/IAAI* (1999).
> > [5] [twitter.com/GaryMarcus/status/1193209251834916864](https://twitter.com/GaryMarcus/status/1193209251834916864)
> > [6] [twitter.com/rasbt/status/1620125103966257152](https://twitter.com/rasbt/status/1620125103966257152)
> > [7] Dziugaite, Gintare Karolina, and Daniel M. Roy. "Data-dependent PAC-Bayes priors via differential privacy." *Advances in neural information processing systems* (2018).
> > [8] Dziugaite, Gintare Karolina, and Daniel M. Roy. "Computing nonvacuous generalization bounds for deep (stochastic) neural networks with many more parameters than training data." *Conference on Uncertainty in Artificial Intelligence* (2017).
> > [9] Maria Perez-Ortiz, Omar Rivasplata, John Shawe-Taylor, and Csaba Szepersvari. “Tighter Risk Certificates for Neural Networks.” *Journal of Machine Learning Research* (2021).
> > [10] Dziugaite, Gintare Karolina, et al. “On the role of data in PAC-Bayes bounds.” *International Conference on Artificial Intelligence and Statistics* (2021).

---

> ### Author Response · Authors · 2023-11-21
> **Following up as the discussion period winds down**
>
> Thank you again for your feedback.  Do you have any other questions?

---

### Official Review · Reviewer_VnMw · 2023-10-28

**Soundness:** 3 good
**Presentation:** 3 good
**Contribution:** 2 fair
**Rating:** 5
**Confidence:** 3

**Summary:**

The authors provide an interpretation of the no-free lunch theorem through the lens of Kolmogorov Complexity and highlight its relations to compressibility. Through this connection, they illustrate that real world datasets are highly structured and they all share some underlying commonalities. No free lunch theorems claim that over the distribution of all possible tasks, every learner will on average perform equally well or worse and the authors claim that while it holds when considering all possible tasks, real world tasks come from a more structured subset of such tasks and are highly compressible, and hence one can obtain reasonable performance over such tasks without having to include task-specific inductive biases as was incentivized through no free lunch theorems.

In particular, the authors use Kolmogorov complexity to derive a new no free lunch theorem, and show using learnability through neural networks that real world datasets can be highly compressible and thus have low complexity. Further, they show that pretrained models like GPT-3 prefer sequences of low complexity, especially even at initialization which is surprising.

**Strengths:**

- The paper is well written and provides a fascinating connection between no free lunch theorems and Kolmogorov complexity. In particular, the approach of using this notion of complexity to show that real world datasets are highly structured is quite interesting.
- The authors also provide a few interesting bounds, especially using the learned $p(y|x)$ distribution to provide a bound on the complexity of the dataset; and then highlighting that as long as the learner achieves better than random chance, it implies that the dataset is compressible.

**Weaknesses:**

- The authors mention that no free lunch theorem holds because when selecting datasets using a uniform distribution, it subtly selects data of high complexity which is not compressible. However, earlier the authors claimed that data of high complexity is exponentially hard to obtain, which means that under uniform distribution over tasks, learners should be able to perform better than random chance because this incompressible data occupies only a small volume. It would be nice if the authors could provide some discussion on this.

- The bound provided in Section 4.1 only requires disproportionate weighting over the hypothesis space based on their complexity. However, it does not take into account the compressible nature of the dataset; and hence it is not clear how the bound is non-trivial for real world datasets when it holds for arbitrary datasets, and in particular could hold for a dataset with high complexity as long as the solution still has low complexity.
- It is not clear what the authors want to show by training a CNN over tabular data. Yes, it is able to perform better than random chance but this is also expected of any, potentially even mismatched, training architecture on any data with some inherent structure. From such a lens, the finding that CNNs are able to generalize on tabular data is not too interesting, so could the authors clarify on why and how it is an interesting finding?
- Details about the GPT model are missing. Are the authors using a pre-trained GPT-3, but if so it has been trained on a lot of different kinds of data and the task of binary tree expansion would be quite OoD for the model. Is it that the authors are training GPT-3 on such a dataset? - It would be nice if the authors actually also do a small scale training (not necessarily GPT-3) and then use that model to provide trends. In particular, the authors could also look at performing training where on average, input of each complexity is seen roughly the same number of times.
- The authors claim that a randomly initialized model could do next token prediction well on low complexity sentences. This is unclear because why would the model be able to do the task at all if it is untrained, irrespective of the complexity of the sentences?
- How do the authors demonstrate or claim that cross validation provably generalizes to millions of models with only thousands of data?
- While it might be okay to use large models with a simplicity preference to provide a universal platform for learning in diverse systems; it is quite unclear how and what the right method of providing simplicity performance to architectures is. For example, in the ViT experiments, the authors could change the simplicity bias by considering the same GoogleLeNet architecture with just $l_2$ penalty, but that does not have as much of an effect as the simplicity bias chosen by authors in Section 5.2.

**Questions:**

*“All but exponentially few sequences of a given length have near maximal Kolmogorov complexity and are thus incompressible…”*
What is the reasoning behind this statement? In the following inequality on probability, is $n$ the size of the bitstring over which a uniform distribution is prescribed? Where does this inequality come from?

*“labels can be encoded in $K(Y|X, p) \leq - \sum_{i=1}^n \log_2 p(y_i | x_i) + 3$ bits”*
How did the authors obtain this inequality?

*“$K(Y | X) \leq K(Y | X, p) + K(p) + 2 \log_2 K(p) + c$”*
How did the authors obtain this inequality?

Unfortunately the proof of Theorem 1 wasn’t clear. Could the authors provide more insight on the statement that there are less than $2^{k+1}$ labelings $Y$ with $K(Y | X) \leq k$.

*“$K_p(h) \leq K(h) + 2 \log_2 K(h)$”*
How did the authors obtain this inequality?

What do the authors mean when they say that the assigned probabilities decay sub-exponentially with sequence length and how does this imply more confidence later on?

*“Randomly initialized models prefer low complexity”*
I am not quite clear on this setting. Given that it is a randomly initialized network, wouldn’t it basically give a uniform distribution over all possible sentences? Even further, the authors claimed that higher complexity sentences are exponentially hard to find, which would imply that the network just gives low probability to complex sentences primarily because they are lower in number. Can the authors provide some of their reasoning on it, and explain how exactly Neural Networks, at initialization, generate something meaningful.

How do the authors demonstrate or claim that cross validation provably generalizes to millions of models with only thousands of data?

---

> ### Author Response · Authors · 2023-11-19
> **Author Response to Reviewer VnMw 1/2**
>
> We thank you for your detailed review and thoughtful feedback. We address your questions and comments below:
>
> > Apparent contradiction between incompressible data being uncommon in the world we observe and common in the uniform distribution
>
> When considering the uniform distribution, high complexity sequences are abundant. In the statement “All but exponentially few sequences …”,  $n$ is indeed the sequence length, and we have now added that context to the paragraph. This classical result follows from the fact that there are at most $2^{L+1}$ terminating programs of length $L$, and therefore only at most that many distinct sequences with complexity $\le L$ out of the total $2^n$ sequences. Therefore the fraction of sequences with length $L$ is less than $2^{1+L-n}$. While high complexity sequences are abundant in the uniform distribution, we argue that they are very uncommon in natural data, including via experiments bounding the Kolmogorov complexity of real-world datasets.
>
> > $2^{k+1}$ labelings Y with $K(Y|X)≤k$
>
> We reuse this classical counting argument for theorem 1. $K(Y|X)$ counts the length of the shortest program that produces $Y$ taking as input $X$. However, there are only at most $\sum_n^k 2^{n} = 2^{k+1}-1 \le 2^{k+1}$ programs of length $\le k$, and therefore there cannot be more than $2^{k+1}$ possible values of $Y$ given a fixed input $X$ (the programs are all deterministic).
>
> **Inequalities:**
> - “labels can be encoded in $K(Y|X, p) \leq - \sum_{i=1}^n \log_2 p(y_i | x_i) + 3$ bits” - This result comes from arithmetic coding using $p(y_i | x_i)$ at each index to form the probability distribution. The size of the arithmetic coding interval is $I = \Pi_i p(y_i|x_i)$ and an interval which is contained in it can be specified in at most $ - \sum_{i=1}^n \log_2 p(y_i | x_i) +2$ (see p128 of Information Theory, Inference, and Learning Algorithms). The +3 is from a suboptimal strategy with additional overhead, but the bound holds true nevertheless.
>
>
> - “$K(Y | X) \leq K(Y | X, p) + K(p) + 2 \log_2 K(p) + c$” - To derive this inequality, we note that $K(Y|X) \le K(Y, p | X)$, $K(Y,p|X) \le K(Y|X) + K(p|X) + 2\log K(p|X)+c$ (see e.g. p3 of [1]), and finally $K(p|X) \le K(p)$.
> - $K_p(ℎ)≤K(ℎ)+2log_{2}⁡K(ℎ)$: this is an upper bound for the cost of converting an arbitrary encoding of $h$ into a prefix free code. Tighter bounds exist, but one way to see the asymptotic behavior is by leveraging arithmetic coding with three symbols ${0,1,STOP}$ according to the corresponding probabilities $1/2-1/(2N),1/2-1/(2N),1/N$, where $N=K(h)$ is the length of the original message. Encoding using arithmetic coding consumes at most $2+N log_2(\frac{2}{1-1/N) + \log_2(1/N)} = N + \log_2\big( \frac{1}{(1-1/N)^N}\big)$ bits, which is less than $N + 2\log_2(N)$, $\forall N \ge 24$.
>
> > “Randomly initialized models prefer low complexity”
>
> While considering a single token at a time (independent of the others) at initialization, there exists a symmetry, so that all token probabilities on average would be as likely as all others. In contrast, it is not the case that a uniform initialization distribution would yield equal probabilities for pairs or sequences of tokens. In fact, it is very biased and this can be observed from the self attention layers:
> $Softmax(QK^T)V = Softmax(X^TW_Q^TW_KX/d)V$. For modern LLMs, the $W_Q$ and $W_K$ matrices are shared, and the random initialization is such that $W_Q^TW_K \approx I$. Therefore the attention at initialization becomes $Softmax(X^TX/d)V$. Considering a single self attention layer, this creates correlations between the tokens that favor similarity, for example.
>
> > How do the authors demonstrate or claim that cross validation provably generalizes to millions of models with only thousands of data?
>
> In Section 5.1, we demonstrate this is the case as an application of finite hypothesis generalization bounds, merely by selecting a uniform prior over the millions of models ($P(h) \ge 10^-8$ for 10 million models) and evaluating the bound with n=20000 for a classification problem with $\delta=.01$. Plugging these values into $R(H) - \hat{R}(h) \le \sqrt{\frac{\log 1/P(h)+\log 1/\delta}{2n}}$, we get $R(H) - \hat{R}(h) \le 0.034$, thereby demonstrating the claim.

---

> > ### Author Response · Authors · 2023-11-19
> > **Author Response to Reviewer VnMw 2/2**
> >
> > **Bounds in section 4.1**
> > These generalization bounds hold regardless of the dataset. For datasets with high complexity, however, it won’t be possible to find a model which simultaneously achieves low risk and low complexity, and eq 1 is actually a result proving that direction. For real world datasets such as image classification or tabular datasets that have much lower than maximal complexity, it **is** possible to use the bounds to nontrivially constrain the model performance, and we demonstrate this in Figure 1 (right).
> >
> > **GPT-3 experiment details**
> > Curiously, despite being OOD as you point out, GPT-3 does not require finetuning data to produce this preference for the low complexity sequences. After generating sequences according to the expression trees, we simply tokenize the digits separately and then feed the resulting token sequence into the model. The details of this experiment are listed in Appendix F.
> >
> > Thank you again for your thoughtful review. We made a significant effort to address your feedback and would appreciate it if you would consider raising your score in light of our response.  Please let us know if you have additional questions we can address.
> >
> > **References:**
> > [1] Lance Fortnow. Kolmogorov complexity. In *Aspects of Complexity (Short Courses in Complexity from the New Zealand Mathematical Research Institute Summer 2000 Meeting, Kaikoura)*, volume 4, pp. 73–86, 2000.

---

> > > ### Comment · Reviewer_VnMw · 2023-11-21
> > > **Official Comment by Reviewer VnMw**
> > >
> > > Thanks to the authors for providing clarifications. All my concerns regarding the theory have been addressed, but I would ask the authors to provide citations and pointers to each inequality that they leverage just to make the draft more readable and complete.
> > >
> > > That being said, I would like to keep my current score as my questions and concerns regarding the experimental setup haven't been addressed. In particular, as far as I know, modern LLMs are trained without sharing of $W_Q$ and $W_K$. Further, if this sharing biases the models to not have a uniform distribution over the space of sequences, can the authors provide ablations showing how this preference changes when sharing or not sharing the weight matrices, as well as some ablations describing how or why such models do not provide a uniform distribution over the space of sequences?
> > >
> > > Additionally, since the authors do not train any models for the expression trees experiment, I am not quite sure what to make of it. Personally, I feel if the authors trained a model on the task of generating such sequences, and then observed this penchant for lower complexity sentences, then it would make a lot more sense than what is being observed currently.
> > >
> > > Further, my concerns regarding the simplicity bias and tabular experiments with CNNs hasn't been addressed. Overall, I feel like it is a very cool work but because of the above reasons, I feel that it is not ready for acceptance in its current form solely based on the relatively weaker experiments section.

---

> ### Author Response · Authors · 2023-11-21
> **Follow up with Reviewer VnMw - 1/2**
>
> Thank you for your engagement, and we are glad that all of your technical questions regarding theory have been addressed.  Furthermore, we have now added citations and pointers to inequalities in Appendix D along with references in the main body of our updated draft, as suggested.  We are grateful for the opportunity to respond to your additional questions below, and we would appreciate it if you could consider updating your score in light of our responses here:
>
> > It is not clear what the authors want to show by training a CNN over tabular data.
>
> These experiments illustrate why a preference for low Kolmogorov complexity alone is sufficient for a nontrivial degree of generalization, provably, since real-world data labelings tend to have low complexity.  We agree a model’s ability to generalize on a modality it was not designed for is not necessarily surprising, but the generalization bounds we compute only make use of the low Kolmogorov complexity preference, and this preference alone is able to explain almost all of the models’ generalization.  Moreover, many in the community, explicitly motivated by no free lunch theorems, do argue that individual applications require specialized learners [1-6], and our work shows why this notion may be misguided in principle. We have now included additional clarifications in our updated draft.
>
> > Unclear how and what the right method of providing simplicity performance to architectures is.
>
> We do not claim to have solved the problem of engineering an optimal low-complexity biased framework, but a point we make is that we do not need to compromise on flexibility in order to express a preference for low complexity solutions.  In the GoogLeNet example, we illustrate the importance of also including the flexibility of a large model since some problems, for example ImageNet, require a higher complexity solution.  Simply put, we want a model that prefers low complexity insofar as low complexity is appropriate for the data we observe; in other words, follow Occam’s Razor and choose the simplest explanation for the training set.  For some datasets, the simplest explanation is more complex than for others.  We have additionally updated the draft to make this point clear.
>
> > The authors do not train any models for the expression trees experiment.
>
> We want to clarify that the examples involving numerical sequences are particularly interesting precisely because we did not train on such sequences.  We show that both randomly initialized LLMs and ones pretrained on a vast text corpus naturally possess a generic notion of low-complexity bias.  These experiments conceptually support the points we make using generalization bounds which explain generalization from an a priori low-complexity bias.  It is also worth pointing out that training GPT-scale LLMs from scratch is infeasible on our computational budget, while a low-complexity bias would be much less surprising and interesting at very small architecture sizes we could afford to train.

---

> ### Author Response · Authors · 2023-11-21
> **Follow up with Reviewer VnMw - 2/2**
>
> **Regarding the non-uniformity of sequences generated by LLMs,** we actually used the stock GPT-2 architecture and initializer off-the-shelf with no modifications at all, which is to say that real architectures in circulation have this behavior.  Moreover, the non-uniformity of labeling functions induced by neural network architectures was previously observed in simple classifiers [7], which we reference in our paper.  Our experiments extend this observation to language models and to notions of complexity which are relevant to our work and notions of learnability from our formulation of the no free lunch theorem. Numerous architectural components (or combinations) may be responsible for this preference for low complexity numerical sequences in practice, including ones which are not yet understood, but our observation is simply that this property exists in neural networks, and it illustrates the low-complexity bias that can explain a nontrivial amount of generalization provably.  Broadly speaking, understanding the components of neural networks which induce their low-complexity bias and how to improve this bias are foundational questions for the success of deep learning, and we intend to make progress on this long-term endeavor in our future work.  As an aside, we suspect that the highly non-uniform distribution over functions induced by a diffuse or uniform distribution over parameters can explain why earlier PAC-Bayes generalization bounds work, even though they did not explicitly incorporate a preference for compressible parameter vectors.  Namely, their diffuse prior probability distributions over parameters implicitly place disproportionate mass on low Kolmogorov complexity functions.
>
> Do you have any other questions we can address?  We appreciate the opportunity to engage with you, we think our draft has benefited from your feedback, and we are happy to provide any other additional information that would be helpful for your evaluation before the discussion phase closes tomorrow.  Our work sheds light on foundational questions which underpin the success of deep learning, and we push back against pervasive and misguided ideas often justified by no free lunch theorems. We think that there is value in re-examining foundational questions and that this work will be broadly interesting to the ICLR community.
>
> **References:**
> [1] Ciuffo, Biagio and Vincenzo Punzo. "“No free lunch” theorems applied to the calibration of traffic simulation models." *IEEE Transactions on Intelligent Transportation Systems* (2013).
> [2] Watson J-P, Barbulescu L, Howe AE, Whitley LD. Algorithm performance and problem structure for flow-shop scheduling. *AAAI/IAAI* (1999).
> [3] Ho, Yu-Chi, and David L. Pepyne. “Simple explanation of the no-free-lunch theorem and its implications.” *Journal of optimization theory and applications* (2002).
> [4] Whitley D, Watson JP. “Complexity theory and the no free lunch theorem.” *Search methodologies* (2005).
> [5] [twitter.com/GaryMarcus/status/1193209251834916864](https://twitter.com/GaryMarcus/status/1193209251834916864)
> [6] [twitter.com/rasbt/status/1620125103966257152](https://twitter.com/rasbt/status/1620125103966257152)
> [7] Valle-Perez, Guillermo, Chico Q. Camargo, and Ard A. Louis. "Deep learning generalizes because the parameter-function map is biased towards simple functions." *International Conference on Learning Representations* (2018).

---

### Official Review · Reviewer_pHWe · 2023-10-31

**Soundness:** 2 fair
**Presentation:** 3 good
**Contribution:** 1 poor
**Rating:** 3
**Confidence:** 4

**Summary:**

**Update after rebuttal:** I thank the authors for their detailed response. Unfortunately most of the criticism raised by me has not led to any changes to the paper, and the authors' response has mainly re-stated their position which is already expressed at length in the paper, but I do not feel that major misunderstandings on my side have been raised or cleared up. This means that my original criticism and assessment stands as is. I still believe that a large part of the material in the paper is so well established that it is found in standard textbooks. I agree with the authors that there is some merit to pointing out these textbook results to the ICLR community, and I must admit that I am quite surprised that some of the other reviewers find it surprising that untrained networks have simplicity biases; after all, there have been dozens of papers over the last years, some of which received considerable attention at top-tier conferences, showing exactly this point (though typically not from the view of low Kolmogorov complexity, but e.g. functional complexity in the frequency domain). Perhaps some of the other reviewers have overstated their self-rated confidence. Regardless, I do not think that the current manuscript constitutes an excellent tutorial or introductory-style paper (which, admittedly, was not the aim of the authors). This leaves the core of the paper being a technical contribution in terms of establishing a "cross-domain generalization bound" and some empirical assessments. This is interesting and valuable, and I personally would much prefer an manuscript that significantly expands on this, presents more rigorous formalism instead of prose, and only devotes at best a paragraph discussing how people commonly misinterpret NFL theorems. In the end it will be for the AC to decide whether the contribution is novel and significant enough, and I will not veto accepting the paper, since I seem to be the outlier among the reviewers. I do strongly agree with the authors that this should be standard knowledge in the community, and I would really like to see more work emphasising the compression viewpoint and how it relates to general learners/learning, and I am sympathetic to their main messages. But stating that neural networks have a bias for low Kolmogorov complexity sequences is a bit too vacuous - this needs to be expanded on (what is the reference machine, or what can we say about what kinds of sequences are simple and complex for neural networks; because clearly, some rather "simple" sequences, like palindromes, are not at all simple under modern neural networks' biases). As it stands, I cannot confidently suggest the manuscript for acceptance at a top-tier conference with a very high bar to pass.

**Detailed comments to the rebuttal:**
 * "This theorem is frequently cited to argue that general purpose learners are impossible in practice, not only in theory.", "Our work argues directly against commonly held beliefs", "Our work fights back against this pervasive idea", "See [1-6] for example claims that the NFL ensures that we need specialized learners for individual applications of machine learning in practice." - I disagree that these are commonly held beliefs; most of the references cited to support this are quite dated or from non-core-ML outlets such as IEEE Transactions on Intelligent Transportation Systems (and I am not engaging with Gary Marcus' tweets here). If anything the last decade in ML has been the overwhelming success of general learners. CNNs replaced hand-crafted, domain-specific features; the end-to-end paradigm took hold, and particularly over the last 3 years transformers are widely hailed as universal learners. I would even go as far and say that never before in the history of ML have more ML researchers believed in the possibility of general learners.
* "In terms of reference machine, this choice is as valid as any other," - ok fine, but it risk sending a bit of a skewed message to readers unfamiliar with Kolmogorov complexity (which is typically understood as being an objective complexity measure, which is watered down by changing the reference machine arbitrarily and/or restricting the set of possible data generators to be non-universal).
* "Rather, we argue that the extent of their low-complexity bias can explain their broad generalization capabilities and why general-purpose machine learning algorithms are possible." This is vacuous, and needs to be stated with more detail (any general-purpose ML algorithm must have, by definition, low-complexity bias; how exactly is the low-complexity bias of neural networks characterized? what kinds of data are low and high complexity under this bias?).
* "study the alignment between the preferences of neural networks and naturally occurring datasets." I really like this point, and think this is where the main contribution of the paper lies. It would be great to expand on this and really (empirically) drill into this to make as concrete statements as possible (what data can neural nets compress easily, what data that we know is easily compressible in principle can neural nets not compress well?).



**Summary:** The paper discusses how No Free Lunch (NFL) theorems rule out a general learner, that is a learning algorithm that works well across all possible datasets. In contrast to the assumptions underlying the NFL theorems, the paper observes that most real-world datasets are compressible, meaning they are structured - or in other words: non-trivial generalization by learning from parts of the data is possible in principle. The question is then whether learning algorithms can be designed that work across many such datasets, which is only possible if the “structure” can be incorporated as a general inductive bias into the learner. The paper proposes to use low Kolmogorov complexity as this structural bias (inspired by Solomonoff induction). After confirming that some commonly used tabular benchmark datasets are indeed compressible by a standard MLP, the paper argues that modern neural networks are candidates for general learners that prefer solutions with low Kolmogorov complexity. This intuition is empirically confirmed on simple computer vision benchmarks (CIFAR, ImageNet) and some synthetic tasks designed such that “Kolmogorov complexity” can be estimated.

**Strengths:**

* Timely and important idea
* Work is situated in a theoretically solid and well understood framework (Solomonoff induction / Kolmogorov complexity
* Paper has polished prose, and nice summaries of the key take-aways of each section
* Simple empirical results that illustrate the main claims

**Weaknesses:**

* By far the biggest weakness of the paper is that its novelty is very limited. Besides the experiments, I would argue that almost all the material can be found in textbooks (which do not cover connections to neural networks) and recent publications (studying inductive biases of neural networks from a complexity perspective). Li & Vitanyi’s ‘An Introduction to Kolmogorov Complexity and Its Applications’ covers a large part of what’s discussed in the paper. I also strongly suggest having a look at Hutter’s ‘Universal Artificial Intelligence’ (which has a brief discussion of NFL vs. Occam’s razor and provides more extensive references on the topic), and Gruenwald’s ‘Minimum Description Length Principle’. To me, the only substantial claim made by the paper that has not been discussed at textbook level before, is that Neural networks (after training, and much more importantly before training) have a bias towards low Kolmogorov complexity patterns. Unfortunately, the paper only scratches the surface in empirically verifying the claim. As a counter-argument consider [1], who investigate neural networks’ ability to learn simple algorithmic patterns across the levels of the Chomsky hierarchy (a hierarchy of computational complexity). Tasks include things like reversing an input string, copying a string, or performing modular arithmetic. The paper finds that standard architectures, in particular transformers fail to generalize on non-regular tasks (i.e. all tasks of higher computational complexity than regular languages). I interpret these results as some low-complexity bias holding on regular-language data, but not on non-regular patterns, e.g. for copying or reversing an input string, very low Kolmogorov complexity programs exist but standard neural networks trained via SGD seem (reliably) not to be able to find them and instead find a solution of higher complexity that does not generalize. These results cannot be reconciled with the broad claim in the paper that neural networks have a general bias for low Kolmogorov complexity. The main claim in the paper is thus either wrong, or somewhat vacuous (in which case it needs further refinement).
* The paper seems to suggest that No Free Lunch theorems somehow rule out the possibility of neural networks that can train successfully on a number of datasets and achieve non-trivial prediction or classification performance. I disagree with this interpretation, and think it is a somewhat common misreading of the NFL theorems. As the paper states, the NFL theorems hold under all possible datasets (or similar formulations depending on the theorem). Clearly the theorems do not apply when considering a subset of all possible datasets, in particular if they share some structure (literally meaning they are compressible, which is dual to saying predictors/classifiers can be trained on a subset of the data and they will generalize). Saying that the NFL theorems do not apply when only considering compressible data is thus at best a tautology, but it does not invalidate the theorems. To me, the NFL theorems are unrelated to this paper since they do not apply in the setting considered by the paper - I personally would drop their discussion from the paper.
* The writing throughout the whole paper is very hand-wavy, and formally well-defined concepts are often used qualitatively and in an imprecise fashion. Just to give one example, Kolmogorov complexity is the length of the shortest program to produce a string **on a universal Turing machine**, and depending on the reference machine the Kolmogorov complexity can change significantly. A programming language can typically be compiled such that it can be executed on a UTM, or it is interpreted by another program running on the UTM. In the limit the length of the compiler/interpreter becomes a negligible constant, but when considering a PyTorch program of 280 characters, the constant is far from negligible. Similarly, Kolmogorov complexity is defined w.r.t. a universal hypothesis class (all computable programs; in a very particular sense); using a more restrictive class like in the experimental tasks defined in the paper, one can construct a complexity measure inspired by Kolmogorov complexity (which does imply losing out on some of the favorable properties of the complexity measure such as universality/”objectivity”); but the numbers reported in the paper should not be called Kolmogorov complexity. Improvement: make all the main concepts introduced (mainly Section 2) precise and formal; add equations. I strongly suggest consulting the corresponding textbooks.

[1] Neural Networks and the Chomsky Hierarchy, Deletang et al. 2022.


**Verdict:**
The topic of identifying the inductive biases of neural architectures (when training via SGD and various regularizers) is timely and important. Recently, a number of papers have investigated the inductive biases of various architectures (CNNs, RNNs, Transformers) either via function-space complexity measures (such as spectral analysis) or information-theoretic / computational complexity measures. The paper, particularly the empirical part, is very much in line with these works, and adds to a body of results that find low-complexity biases in standard architectures / training setups. Admittedly, the background discussed in the paper (universal induction and low Kolmogorov complexity, or dually, compressibility as a general inductive bias) does perhaps not receive enough attention in the wider ML community recently, and a good tutorial/introduction could be a valuable contribution in itself. Unfortunately the current version of the paper falls short of providing a concise yet formally precise introduction / tutorial by being too hand-wavy throughout (also I believe that such a tutorial would require at least the full 9 pages, maybe a bit more, making it less suitable as a conference paper). Overall I could see this work being developed in two ways: double down on the tutorial aspect (highlighting important well-established theoretical results and situating them within modern ML practice), or double down on the empirical aspect (are there classes of data generators where the low Kolmogorov complexity bias does not hold for standard architectures, or can we really claim universality of the findings in the paper; see my earlier comment on at least one strong counter example with non-regular languages)? If either of these lines were fully developed, I believe that would constitute a solid and interesting contribution, but I personally think the current manuscript needs more work before it is ready for publication. I am happy to read the other reviews and hear the authors’ response before coming to a final conclusion.

**Questions:**

**Comments:**

* I disagree with the premise that no-free lunch theorems state that “the world is hostile to inductive reasoning”. Obviously we do not live in a world that presents us with a uniform distribution over all possible learning problems. To make a philosophical point, since evolution is also bound by the limits of computability and compressibility, it is expected that life can only thrive in compressible (=predictable) environments - we do also encounter (virtually) incompressible data, but we typically deem it irrelevant, a.k.a. noise; we (can) only care about the compressible aspects of the world we live in.
* End of 3.1: “allowing us to reject the hypothesis that the labeling functions are drawn uniformly at random with extremely high confidence.” Who would make the claim that labeling functions for real-world datasets are drawn uniformly at random?
* Section 5: “Whereas the no free lunch theorems seemingly preclude automated meta-learners which select performant models on any task, [...] the defeating conclusion of Wolpert’s no free lunch theorem is escaped as long as datasets share structure so that the model selector generalizes to new datasets.” Who would claim otherwise? I would be willing to make a significant bet that Wolpert would strongly agree with the sentence in the paper. Note how the first part of the sentence (in line with the NFL theorems) says **any task**, but the second part of the sentence only considers **datasets that share structure**. These two kinds of settings are not fundamentally different; due to the restriction in the second part of the sentence the claim does not invalidate or nullify the NFL theorems.
* Section 5.1: “A near state-of-the-art computer vision model can be expressed in only 280 characters [...] in PyTorch.”. Yes, but this does not imply that the Kolmogrov complexity is 280 times(!) 8 bits. In the regime of such short programs the reference machine (and thus all libraries and the compiler/interpreter) must be taken into account too, pushing the (upper bound) of the Kolmogorov complexity into the tens of thousands of bits.

---

> ### Author Response · Authors · 2023-11-19
> **Author Response to Reviewer pHWe 1/3**
>
> Thank you for your feedback and for recognizing the importance, timeliness, and simplicity of our work.  We address each of your points below:
>
> > Its novelty is very limited.
>
> We indeed cite a number of these works on Kolmogorov complexity, universal induction in the context of no free lunch theorems, or simplicity bias in neural networks.  However, no free lunch theorems are commonly referenced in support of the need for domain-specific learners, and the community lacks an understanding of why machine learning algorithms possess generic generalization capabilities across real-world problems. See [1-6] for example claims that the NFL ensures that we need specialized learners for individual applications of machine learning in practice.
>
> In our work, we synthesize an argument for shared common structure of real-world data distributions and neural networks, including numerous novel experiments from computing the first cross-domain generalization bounds to measuring the preference of LLMs for low-complexity numerical sequences, and we relate this phenomenon to NFL by deriving a new no free lunch theorem in terms of compressibility.  Our formulation of the no free lunch theorem allows us to tease out the applicability of no free lunch theorems to real-world problems by performing compression of datasets and models.  Our work argues directly against commonly held beliefs, and we synthesize a story that to our knowledge does not exist in its full form in the literature.  Moreover, our experiments lead to conclusions which will likely be surprising to many in the community.  For example, Reviewer VnMw says "given that it is a randomly initialized network, wouldn’t it basically give a uniform distribution over all possible sentences?"  In contrast to the reviewer's intuition, we show that randomly initialized LLMs induce a highly non-uniform distribution over sequences, namely one which puts disproportionate mass on low-complexity sequences.  Due to the controversial nature of our overarching argument, our novel experiments, and formulation of the no free lunch theorem, this work will attract attention and contribute significantly to ICLR 2024.
>
> > As a counter-argument consider [1], who investigate neural networks’ ability to learn simple algorithmic patterns across the levels of the Chomsky hierarchy (a hierarchy of computational complexity).
>
> The point of our paper is not that neural networks perfectly mimic Solomonov induction or are near-optimal in any way, from the viewpoint of low-complexity bias. Rather, we argue that the extent of their low-complexity bias can explain their broad generalization capabilities and why general-purpose machine learning algorithms are possible.  We don’t disagree with the point that in practice a given function may be hard to learn for a network even if it can be expressed in the model weights. Furthermore, many model architectures like transformers don’t have the ability to express an algorithm that solves a given problem due to the complexity class and how the compute grows as a function of the context length.  However, we don’t think that the imperfections of neural networks detracts from the arguments we make in our paper.

---

> ### Author Response · Authors · 2023-11-19
> **Author Response to Reviewer pHWe 2/3**
>
> > The paper seems to suggest that No Free Lunch theorems somehow rule out the possibility of neural networks that can train successfully on a number of datasets and achieve non-trivial prediction or classification performance … I disagree with the premise that no-free lunch theorems state that “the world is hostile to inductive reasoning”.
>
> The No Free Lunch Theorem of Wolpert (1996) proves the non-existence of general purpose learners and the hostility of the world to induction under the very strong assumption that learning problems are uniformly distributed.  This theorem is frequently cited to argue that general purpose learners are impossible in practice, not only in theory.  For example, [1] say “the performance of the different algorithms over the 42 experiments considerably differ. This proves the validity of NFL theorems in our field”,  [2] reference NFL to explain why success on synthetic problems may not transfer to real ones, [3] say “the only way one strategy can outperform another is if it is specialized to the specific problem under consideration”, and [4] reference the NFL in stating that “the business of developing algorithms is one of building special-purpose methods to solve application-specific problems”.  See [5,6] for examples of prominent community members leveraging the NFL to argue for specialized learners.
>
> Our work fights back against this pervasive idea, arguing that real-world data shares a tremendous amount of structure in common with neural networks.  We formalize the no free lunch problem in terms of compression, delineating between the compressible regime and incompressible regime where learning is impossible.  We then examine the extent to which neural networks can compress data.  Our experiments include results which fly in the face of arguments for specialized learners, for example we demonstrate the ability of convolutional networks to compress tabular data (with features written as pixels) which lacks any spatial structure and consequently prove highly non-vacuous cross-domain PAC-Bayes generalization bounds.  In summary, while we agree that real-world learning problems are clearly non-uniform, there is a controversy surrounding the extent to which learning problems share common structure and the extent to which general purpose learners are possible.  Our work synthesizes an explanation for why general purpose learners are possible and presents numerous novel experiments, including ones which show how established ideas, for example from information theory or learning theory, can explain the abilities of modern neural networks including LLMs.
>
> > The writing throughout the whole paper is very hand-wavy... A programming language can typically be compiled such that it can be executed on a UTM, …, but when considering a PyTorch program of 280 characters, the constant is far from negligible.
>
> Your point that for very low complexity sequences, Kolmogorov complexity can vary significantly with different choices of the reference machine is well noted, and this choice (equivalent to a choice of language L) is referenced in our background section on Kolmogorov complexity. In this case, we define L using the Python programming language equipped with PyTorch and the standard libraries (and if desired the code to compile these libraries to any other reference turing machine). In terms of reference machine, this choice is as valid as any other, and the complexities computed (with the exception of the expression tree experiments) are with respect to this reference machine.
>
> > Using a more restrictive class like in the experimental tasks defined in the paper, ... the numbers reported in the paper should not be called Kolmogorov complexity
>
> In Section 4.3, we consider a simplified non-universal language using expression trees. Language is included as a subset of the universal hypothesis class. If sequences are simple in this simplified language, then with some number of additional bits to encode the description of this language, the sequences will be simple according to Kolmogorov complexity too. However, we take your point that the language in this paragraph could be made more precise, and we have done so in the updated draft.

---

> > ### Author Response · Authors · 2023-11-19
> > **Author Response to Reviewer pHWe 3/3**
> >
> > > End of 3.1: “allowing us to reject the hypothesis that the labeling functions are drawn uniformly at random with extremely high confidence.”
> >
> > Section 3.1 shows how we can view neural networks as data compressors and how we can use this idea to study the alignment between the preferences of neural networks and naturally occurring datasets.  In Section 3.2, we present a no free lunch theorem in terms of compressibility.  Combining our theorem with the compression experiments illustrates how neural networks avoid the incompressibility regime where learning is impossible.  Broadly, we argue that the structure in real-world data enables general purpose learners.  For example, we present novel cross-domain generalization bounds for convolutional networks, designed for data with spatial structure, applied to tabular data which lacks any spatial structure.  This view that real world data distributions share structure, in fact enough to enable general-purpose learners, is actually a highly controversial position.  See above for a discussion about this controversy and the common mis-use of no free lunch theorems.
> >
> > > Note how the first part of the sentence (in line with the NFL theorems) says any task, but the second part of the sentence only considers datasets that share structure… the claim does not invalidate or nullify the NFL theorems.
> >
> > To clarify, we absolutely do not claim that the NFL theorems are false or that their proofs contain errors.  They are commonly invoked to support the claim that general-purpose learners are impossible.  We argue that real-world data distributions in fact share enough common structure, and this structure is reflected by neural networks, so that a single model can solve a wide range of the types of problems we see in the real world.  In other words, while a single learner can’t solve any task as per NFL, it may be able to solve any task we wish to solve in the real world.  We synthesize this story through the lens of compression, including novel experiments and a novel no free lunch theorem.  We stress that while you seem to agree with the notion that no free lunch theorems are not limiting in practice, this view is controversial, and we feel that our work will be interesting to the broad ICLR community.
> >
> > Thank you again for your thoughtful review. We made a significant effort to address your feedback and would appreciate it if you would consider raising your score in light of our response.  Please let us know if you have additional questions we can address.
> >
> > **References:**
> > [1] Ciuffo, Biagio and Vincenzo Punzo. "“No free lunch” theorems applied to the calibration of traffic simulation models." *IEEE Transactions on Intelligent Transportation Systems* (2013).
> > [2] Watson J-P, Barbulescu L, Howe AE, Whitley LD. Algorithm performance and problem structure for flow-shop scheduling. *AAAI/IAAI* (1999).
> > [3] Ho, Yu-Chi, and David L. Pepyne. “Simple explanation of the no-free-lunch theorem and its implications.” *Journal of optimization theory and applications* (2002).
> > [4] Whitley D, Watson JP. “Complexity theory and the no free lunch theorem.” *Search methodologies* (2005).
> > [5] [twitter.com/GaryMarcus/status/1193209251834916864](https://twitter.com/GaryMarcus/status/1193209251834916864)
> > [6] [twitter.com/rasbt/status/1620125103966257152](https://twitter.com/rasbt/status/1620125103966257152)

---

> ### Author Response · Authors · 2023-11-21
> **Following up as the discussion period winds down**
>
> Thank you again for your feedback.  Do you have any other questions?

---

### Official Review · Reviewer_wTgY · 2023-11-06

**Soundness:** 3 good
**Presentation:** 2 fair
**Contribution:** 2 fair
**Rating:** 6
**Confidence:** 2

**Summary:**

The paper challenges the No Free Lunch theorems' notion that every learning problem requires a unique algorithm. It argues that neural networks inherently prefer low-complexity data, which is common in real-world scenarios, and can generalize across various domains. The research shows that neural networks can compress data and generate low-complexity sequences effectively, even with little customization. This suggests the possibility of developing universal learning algorithms, thus supporting the trend towards more generalizable and fewer machine learning models in deep learning.

**Strengths:**

The paper presents an original reexamination of the No Free Lunch theorems, offering a fresh perspective by proposing that neural networks inherently favor low-complexity data, challenging the belief that each learning problem requires a distinct algorithm. The quality of work is evident as it is well-founded on theoretical bases and is further bolstered by empirical experiments. It is written with commendable clarity, managing to articulate the interplay between complex theoretical concepts and their practical applications in machine learning. Its significance is underscored by its potential to influence the development of generalized learning algorithms, marking a substantial leap in the field's evolution towards more efficient and versatile machine learning models. This paper stands out for bridging abstract theory with concrete experimental evidence, making a significant contribution to the literature.

**Weaknesses:**

1. The assertion that a universal high-degree polynomial can be effectively applied across diverse sample sizes, given a bias towards simplicity, lacks comparative analysis with contemporary leading methods.

2. The commentary on the combined effectiveness of GoogLeNet and ViT, underpinned by a simplicity preference, does not evidently align with the primary findings of the study.

3. The authors propose a heuristic for addressing the non-computability of Kolmogorov complexity by employing a simplified, non-universal computational language. However, the adequacy of this approximation, both theoretically and empirically, remains unexplained.

4. While the findings of the paper are intriguing, many of the experimental results seem tangential or conceptually disconnected from the core conclusions drawn in Theorem 1.

**Questions:**

In addressing the non-computability of Kolmogorov complexity, the authors choose a simplified language as a heuristic. Could they elaborate on the justification for this choice and its validity as a proxy, both theoretically and empirically?

---

> ### Author Response · Authors · 2023-11-19
> **Author Response to Reviewer wTgY**
>
> Thank you for your supportive feedback and for recognizing the originality, significance, and quality of our work.  We address your individual comments below:
>
> 1.  "The assertion that a universal high-degree polynomial can be effectively applied across diverse sample sizes, given a bias towards simplicity, lacks comparative analysis with contemporary leading methods."
>
> Our polynomial experiment serves as an illustration of a broader principle, namely that the low complexity of datasets allows a single model to perform well on many problems (in this illustration, sample sizes) simultaneously as long as the model too encodes a preference for low complexity.  This illustration is not intended to advocate for polynomials in particular as a competitive method for machine learning problems  In fact, other expressive models likely encode an even better approximation of the Solomonov prior, for example, on relevant problems.
>
> 2.  "The commentary on the combined effectiveness of GoogLeNet and ViT, underpinned by a simplicity preference, does not evidently align with the primary findings of the study."
>
> This experiment shows that problems which seemingly benefit from different learners (e.g. small datasets vs large datasets) can be solved by a single learner with a simplicity bias.  This point aligns with the broad message of our paper, namely that no free lunch theorems do not limit machine learning since real data tends to share common structure, enabling general purpose learners in principle.  On the other hand, this experiment also verifies that ViTs do not possess as strong a simplicity bias as GoogLeNet, highlighting that different neural networks exhibit varying degrees of simplicity bias, and this bias can affect performance on various tasks.  Thank you for pointing this out, and we have updated our draft with a corresponding discussion.
>
> 3.  "The authors propose a heuristic for addressing the non-computability of Kolmogorov complexity by employing a simplified, non-universal computational language."
>
> We adopt three different setups for complexity: (a) we bound Kolmogorov complexity directly via compression.  In an ideal world, we could use Kolmogorov complexity across the boards, but comparing Kolmogorov complexity of different sequences is infeasible because we can only compare upper bounds; (b) we employ expression trees in Section 4.3. This setup allows us to measure complexity exactly for small sequences, and it allows us to bound Kolmogorov complexity formally as we discuss, but the language has the drawback of being simpler than a universal language; (c) we generate periodic sequences in 4.4.  This language is even less rich than the expression trees, but it enables us to measure the complexity of very long sequences which are useful for experiments with randomly initialized models.  Nonetheless, this third setup also is conveniently related to an upper bound on Kolmogorov complexity (e.g. in Python) given by the length of say print(n*x), where n denotes the number of repetitions, and x denotes the repeated subsequence.
>
> 4.  "While the findings of the paper are intriguing, many of the experimental results seem tangential or conceptually disconnected from the core conclusions drawn in Theorem 1."
>
> Theorem 1 delineates the cases where data is highly structured and those where data is incompressible and learning is impossible.  We then spend the remainder of the paper showing that both real-world data and ML models share a common notion of structure and hence why learning is possible in practice.  Proving a no free lunch theorem in terms of compressibility allows us to explore the extent to which that theorem is limiting in the real world by applying compression to datasets and models.
>
> Thank you again for your thoughtful review. We made a significant effort to address your feedback and would appreciate it if you would consider raising your score in light of our response.  Please let us know if you have additional questions we can address.

---

> ### Author Response · Authors · 2023-11-21
> **Following up as the discussion period winds down**
>
> Thank you again for your feedback.  Do you have any other questions?

---

### Meta-Review · Area_Chair_3GJB · 2023-12-11

**Metareview:**

This submission surveys the notion of inductive biases in machine learning, arguing that the underlying reason that standard architectures are transferable across tasks and modalities is due to some universality properties in the distribution of natural data. However, what precisely these properties are is not advanced in this work. Moreover, I agree with several reviewers that these notions (compression & generalization; natural distributions are not all distributions) are common knowledge in machine learning, thus putting this paper in a position that is difficult to advocate for as a clear advancement of the discussion of the role of inductive bias. In its current form, this manuscript is better suited for a venue that accepts survey or position papers.

**Justification For Why Not Higher Score:**

surveys but does not advance the discussion on inductive biases in standard machine learning pipelines

**Justification For Why Not Lower Score:**

N/A

---

### Decision · Program_Chairs · 2024-01-16

Reject